# Beyond Multi-token Prediction: Pretraining LLMs with Future Summaries

**Divyat Mahajan**[1,*]  **Sachin Goyal**[2]  **Badr Youbi Idrissi**[3]  **Mohammad Pezeshki**[3]
**Ioannis Mitliagkas**[1]  **David Lopez-Paz**[3]  **Kartik Ahuja**[3]

[1]Mila, Université de Montréal   [2]Carnegie Mellon University   [3]FAIR at Meta

## Abstract

Next-token prediction (NTP) has driven the success of large language models (LLMs), but it struggles with long-horizon reasoning, planning, and creative writing, with these limitations largely attributed to teacher-forced training. Multi-token prediction (MTP) partially mitigates these issues by predicting several future tokens at once, but it mostly captures short-range dependencies and offers limited improvement. We propose future summary prediction (FSP), which trains an auxiliary head to predict a compact representation of the long-term future, preserving information relevant for long-form generations. We explore two variants of FSP: *handcrafted summaries*, for example, a bag of words summary of the future of the sequence, and *learned summaries*, which use embeddings produced by a reverse language model trained from right to left. Large-scale pretraining experiments (3B and 8B-parameter models) demonstrate that FSP provides improvements over both NTP and MTP across math, reasoning, and coding benchmarks.

## 1 Introduction

Early progress in large language models was primarily driven by massive scaling of both data and compute (Brown et al., 2020; Kaplan et al., 2020). However, the returns from this vanilla scaling approach are beginning to diminish as we encounter the "data wall" (Sutskever, 2024). This has renewed efforts toward algorithmic advances, including new architectures and pretraining objectives, that can extract more predictive signal from a fixed amount of training data.

Next-token prediction (NTP) with teacher forcing—training models by conditioning on ground-truth history when predicting the next token—is foundational to current pretraining methods. However, this approach introduces a train-inference mismatch known as exposure bias: during inference, the model must rely on its own outputs rather than the ground truth, leading to compounding errors and degraded long-range generation quality (Bengio et al., 2015). Moreover, teacher forcing can induce training-time shortcut learning as well, where the model exploits local cues from the ground-truth prefix instead of capturing true long-range dependencies (Bachmann & Nagarajan, 2024). These issues manifest most clearly in tasks demanding extended reasoning, narrative coherence, and open-ended creativity (Papalampidi et al., 2022; Nagarajan et al., 2025).

An appealing alternative to teacher forcing is teacherless training, where the model learns from its own generations rather than relying on ground-truth histories. However, this approach is computationally intensive and challenging to parallelize. As a practical compromise, recent research has explored multi-token prediction (MTP) methods (Gloeckle et al., 2024), which train auxiliary heads to predict several future tokens simultaneously. MTP has demonstrated success in large-scale systems such as DeepSeek-V3 (Liu et al., 2024) and Qwen-3 (Yang et al., 2025). In MTP, each time step augments the standard next-token prediction ($x_{t+1}$) with auxiliary heads that predict additional future tokens (e.g., $x_{t+2}$ and beyond). However, these methods typically assume independence among the predicted tokens given the prefix, resulting in poor approximations of the true joint distribution over long future spans.

---

*Correspondence to: `divyatmahajan@gmail.com`. Work done during DM and SG's internship at Meta.

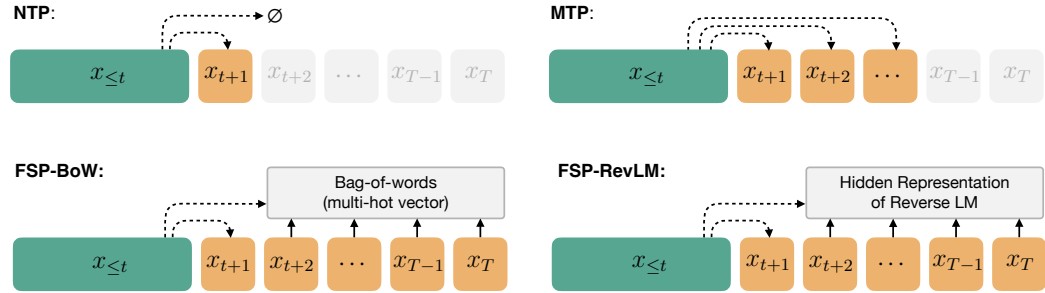

Figure 1: *A comparison of future-aware pretraining objectives.* All methods take a prefix $x_{\leq t}$ as input. **NTP**: Only predicts the immediate next token. **MTP**: Uses multiple auxiliary heads, each predicting a specific future token. **FSP-BoW**: Our proposed hand-crafted summary method that predicts a multi-hot "bag-of-tokens" summary of a long future window using a single auxiliary head. **FSP-RevLM**: Our proposed learned summary method predicts a compact hidden representation of the future, which is generated by a reverse language model (RevLM).

In this work, we propose a different approach: rather than predicting multiple future tokens individually, we train a *single* auxiliary head to *predict a summary representation of the future sequence.* It aims to push as much information as possible about the future into a single target vector, while filtering out information that is inherently unpredictable. At each time step $t$, given the future tokens $(x_{t+2}, x_{t+3}, \ldots, x_{t+\tau})$, we construct a summary vector $a(t, \tau)$ to supervise the auxiliary head. We explore two complementary approaches for constructing future summaries: a simple token-level aggregation method, and our main contribution that learns a representation of the future sequence.

- **Hand-crafted summary.** We train the auxiliary head to predict *all* the future tokens that will occur in a future window, without requiring to know their exact positions. Concretely, at each step $t$, we define a multi-hot vector $a(t, \tau)$ over the vocabulary, akin to bag-of-words, where $a(t, \tau)_i = \mathbb{I}(i \in \{x_{t+2}, \ldots, x_{t+\tau}\})$, and train the model with a binary cross-entropy objective.
- **Learned summary.** Handcrafted summaries such as the one proposed above can be noisy as not all future tokens are equally relevant. To address this, we propose to *learn* a compact summary of the future. We do this by training a reverse language model ($g_r$) on reversed sequences, so that its hidden representation $a(t, \tau) = g_r(x_{\geq t+2})$ serves as a rich embedding of the future context. The auxiliary head of the forward model is then trained to match this representation with an $\ell_2$ loss.

We use the lens of synthetic tasks, path-star graph and sibling discovery (Bachmann & Nagarajan, 2024; Nagarajan et al., 2025) to clarify the conceptual difference between MTP and our approach. On path-star graph, hand-crafted summaries deliver strong gains over MTP, which only predicts immediate future tokens, demonstrating the benefit of long-horizon supervision. However, in sibling discovery, where the future includes content unrelated to the current local sequence, handcrafted summaries struggle, as they treat all future tokens equally. In contrast, our learned summary vectors focus on the informative parts of the future and achieve consistent gains.

We then scale to real-world pretraining at the 8B parameter level, conducting a systematic evaluation across six methods, including DeepSeek-MTP and multiple handcrafted summary variants—a breadth of comparison that is rare due to its high computational cost. The proposed future summary prediction with learned summaries yields substantial improvements over NTP and MTP baselines, with gains of up to 5% on math and coding benchmarks that demand long-horizon reasoning and planning. These results demonstrate that future summary prediction is not only effective in controlled synthetic settings but also translates into meaningful gains for large-scale LLM training.

## 2 FUTURE-AWARE PRETRAINING

### 2.1 BACKGROUND ON NEXT-TOKEN AND MULTI-TOKEN PREDICTION

Let define a sequence of tokens as $X = (X_1, X_2, \ldots, X_T)$ sampled from a distribution $\mathbb{P}_X$, where each token is a discrete random variable supported on $\{1, 2, \ldots, V\}$.

**Next-token Prediction (NTP).** The standard next-token prediction objective is:

$$L_{\text{NTP}}(X, P_\theta) = -\mathbb{E}_{x \sim \mathbb{P}_X} \left[ \sum_{t=1}^{T-1} \log P_\theta(x_{t+1} \mid x_{\leq t}) \right], \tag{1}$$

where $P_\theta$ is the predictor, optimized via

$$\theta^\star = \arg\min_\theta L_{\text{NTP}}(X, P_\theta). \tag{2}$$

A typical parameterization is:

$$P_\theta(x_{t+1} \mid x_{\leq t}) = \text{softmax}\Big( f_u \circ f_h \circ f_s (x_{\leq t}) \Big), \tag{3}$$

where $f_s$ is the transformer backbone, $f_h$ is a processing head, and $f_u$ is the unembedding layer (Figure 2).

**Multi-token Prediction (MTP).** MTP aims to jointly predict multiple future tokens, as follows:

$$L_{\text{MTP-Joint}}(X, P_\theta) = -\mathbb{E}_{x \sim \mathbb{P}_X} \left[ \sum_{t=1}^{T-1} \log P_\theta(x_{t+1}, \ldots, x_{t+\tau} \mid x_{\leq t}) \right]. \tag{4}$$

Since modeling the exact joint distribution is intractable, a common simplification is to model the marginal distribution of future tokens given the prefix (Gloeckle et al., 2024; Liu et al., 2024):

$$L_{\text{MTP}}(X, P_\theta) = -\mathbb{E}_{x \sim \mathbb{P}_X} \left[ \sum_{t=1}^{T-1} \sum_{k=1}^{\tau} \mathbf{1}[t + k \leq T] \log P_\theta(x_{t+k} \mid x_{\leq t}) \right]. \tag{5}$$

Following Gloeckle et al. (2024), the predictor uses separate auxiliary heads for each $k$:

$$\begin{aligned} P_\theta(x_{t+1} \mid x_{\leq t}) &= \text{softmax}\Big( f_u \circ f_h \circ f_s (x_{\leq t}) \Big), \\ P_\theta(x_{t+k} \mid x_{\leq t}) &= \text{softmax}\Big( f_u \circ f'_{h_k} \circ f_s (x_{\leq t}) \Big), \quad \forall k > 1, \end{aligned} \tag{6}$$

where $f'_{h_k}$ are auxiliary transformer blocks specialized for predicting future tokens $x_{t+k}$.

This design reduces teacher forcing by predicting $x_{t+k}$ from $x_{\leq t}$ only, rather than conditioning on the full prefix $x_{\leq t+k-1}$. This is the key principle behind multi-token prediction, i.e., reduced teacher forcing by requiring the model to predict a block of future tokens at each step.

We discuss other variants of MTP such as DeepSeek-MTP and random future token MTP (Thankaraj et al., 2025) in Appendix B.

## 2.2 FUTURE SUMMARY PREDICTION (FSP)

In MTP, we use a set of auxiliary heads to predict a block of immediate future tokens. A key limitation of this approach is that one does not know exactly where the informative signal in the future sequence lies. Informative signals in future could occur far away in the sequence and be well beyond the number of future tokens ($k$) that MTP predicts. A trivial approach to overcome this could be to predict all the future tokens. However, having one auxiliary head per future token is not scalable, and limits the amount of future tokens we can utilize during training.

Towards this, we propose *Future Summary Prediction* (FSP), that predicts a compact summary of the (long) future sequence rather than each token individually.

Let $a(t, \tau)$ represent a summary of future tokens $(x_{t+2}, \ldots, x_{t+\tau})$. The learning objective is:

$$L_{\text{FSP}}(X, P_\theta) = L_{\text{NTP}}(X, P_\theta) + \mathbb{E}_{x \sim \mathbb{P}_X} \Big[ l_a \big( A_\phi(x_{\leq t}), a(t, \tau) \big) \Big], \tag{7}$$

where $P_\theta$ is the next-token predictor, $A_\phi$ is the summary predictor, and $l_a$ is the loss between predicted and ground-truth summaries. The architecture is:

$$\begin{aligned} P_\theta(x_{t+1} \mid x_{\leq t}) &= \text{softmax}\Big( f_u \circ f_h \circ f_s (x_{\leq t}) \Big), \\ A_\phi(x_{\leq t}) &= f'_{h_a} \circ f_s (x_{\leq t}). \end{aligned} \tag{8}$$

Unlike MTP, FSP requires only a single auxiliary head $f'_{h_a}$, making it more scalable.

The key question, then, is how to construct an effective future summary. In this work, we investigate two approaches:

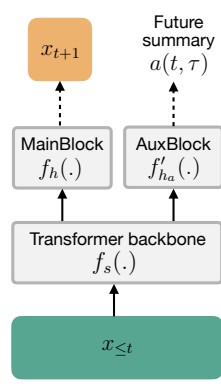

Figure 2: An abstraction of the architecture that subsumes NTP, MTP, and FSP.

**(1) Hand-crafted future summaries.** We define a binary vector $a(t, \tau) \in \mathbb{R}^V$ indicating whether token $i$ appears in the future window:

$$a(t, \tau)_i = \mathbb{I}\big(i \in \{x_{t+2}, \ldots, x_{t+\tau}\}\big). \tag{9}$$

Given logits from $A_\phi(x_{\leq t})$, we minimize a reweighted binary cross-entropy loss:

$$l_a\big(A_\phi(x_{\leq t}), a(t, \tau)\big) = -\sum_{i=1}^{V} w(i)\Big[a_i \log \sigma(z_i) + (1 - a_i) \log\big(1 - \sigma(z_i)\big)\Big] \tag{10}$$

where $a_i = \mathbb{I}\big(i \in \{x_{t+2}, \ldots, x_{t+\tau}\}\big)$, $z_i$ is the $i$-th logit of $A_\phi(x_{\leq t})$, $\sigma$ is the sigmoid, and $w(i)$ reflects the importance of token $i$ (e.g. term frequency-inverse document frequency, tf–idf).

**(2) Learned future summaries.** Predicting every future token via hand-crafted summaries as discussed above could be noisy. For example, not all tokens in the future are informative and thus hand-crafted summaries that account for all of them can be wasteful. Instead of predicting all the future tokens, we therefore propose to predict a learned representation of the future tokens relevant to predicting the current token $x_{t+1}$. We learn this representation via a reverse language model $Q_\psi$, (RevLM), trained on "right-to-left" sequences:

$$L_{\text{RevLM}}(X, Q_\psi) = -\mathbb{E}_{x \sim \mathbb{P}_X}\left[\sum_{t=1}^{T-1} \log Q_\psi(x_{t+1} \mid x_{\geq t+2})\right]. \tag{11}$$

RevLM shares the same architecture, and we take its hidden state as the summary vector:

$$a(t, T - t) = g_h \circ g_s\big(x_{\geq t+2}\big). \tag{12}$$

We then train $A_\phi(x_{\leq t})$ to match this representation via the $\ell_2$ loss:

$$l_a\big(A_\phi(x_{\leq t}), a(t, T - t)\big) = \big\|A_\phi(x_{\leq t}) - g_h \circ g_s(x_{\geq t+2})\big\|_2^2. \tag{13}$$

**Why Future Summary Prediction?** The key advantage of future summary prediction is its ability to reduce dependence on teacher forcing when modeling long future sequences. To intuitively measure teacher forcing, consider for each ground-truth token exposed to the model, how much information is the model required to predict about unseen tokens? If the model predicts more such information, then we have reduced teacher forcing. Next-token prediction (NTP) uses the highest degree of teacher forcing, since the model always conditions on ground-truth histories to predict just the next token. Multi-token prediction (MTP) partially relaxes this by asking the model to predict short blocks of future tokens, thereby reducing teacher forcing locally. However, MTP remains constrained by the short horizon of its predictions. In contrast, our proposed approach predicts summaries of long future sequences, substantially reducing teacher forcing by requiring the model to reason about rich, global properties of the target trajectory.

## 3 ANALYZING FUTURE SUMMARIES

Future summary prediction provides a unified view of a broad family of pretraining objectives, covering MTP and its variants that sample random future tokens (Thankaraj et al., 2025; Gerontopoulos et al., 2025), and our proposed bag-of-words (FSP-BoW) and ReverseLM (FSP-RevLM) summary prediction objectives.

To better understand how the FSP framework clarifies the tradeoffs of each objective on long-horizon planning, we consider the graph modeling benchmarks introduced in prior works. Our analysis yields following insights:

- **Long future summaries matter.** On the canonical path–star task (Bachmann & Nagarajan, 2024), we find that MTP with short range future prediction fails to generalize, highlighting the need for auxiliary objectives that incorporate *long-range* future information.

- **Adaptive future summaries matter.** On a modified sibling discovery task (Nagarajan et al., 2025), we show that incorporating every future token with hand-crafted summaries is suboptimal, highlighting the need for *learned* summaries that retain key information.

## 3.1 Long future summary is important

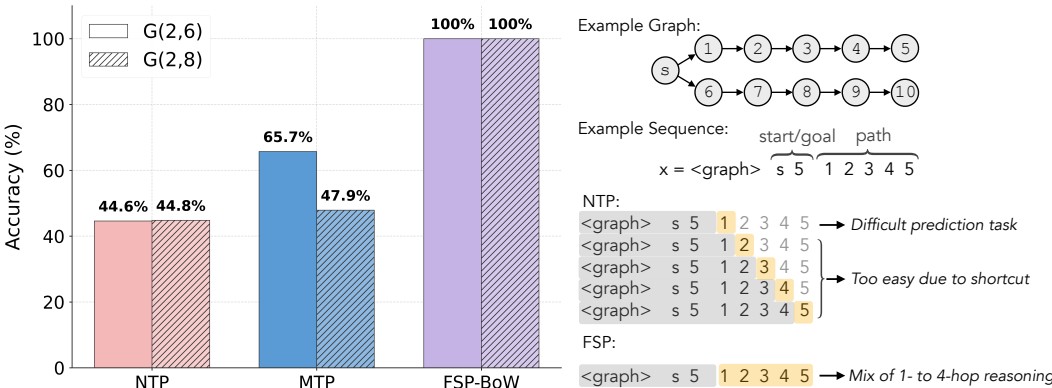

Figure 3: *Analysis of FSP-BoW on the path-star task, which tests long-horizon planning.* **Left:** Accuracy (mean over 5 random seeds) of different pretraining objectives on degree 2 graphs with path lengths 6 and 8. Standard NTP generalizes poorly, and MTP's accuracy degrades as the path length increases, while FSP-BoW achieves perfect accuracy. **Right:** An illustration of why NTP fails while FSP-BoW succeeds, where the input context is shown in `grey` and the target is in `beige`.

We consider the path-star graph, a directed acyclic graph (DAG) $G(d, l)$ composed of $d$ paths, each of length $l$, originating from a central start node $v_{\text{start}}$. The model is given the adjacency list of the graph in the prefix, and the task is to generate the path from $v_{\text{start}}$ to a designated end node $v_{\text{end}}$.

Let the target path be $(v_{\text{start}}, v_1, v_2, \cdots, v_{\text{end}})$ and the input prefix be $p = (Adj(G), v_{\text{start}}, v_{\text{end}})$. Then NTP with teacher forcing predicts an intermediate node in the path as $P_\theta(v_{i+1}|p, v_{\leq i})$. As shown by Bachmann & Nagarajan (2024), NTP often learns shortcut solutions: the model can recover $v_{i+1}$ directly from $v_i$ by scanning the adjacency list in $p$, without learning the underlying long-range plan (Figure 3). This leads to gradient starvation (Pezeshki et al., 2021), where the supervision signal for the actual planning task is lost. Once the shortcut is learned, meaningful gradient information remains only for predicting the first step $v_1$, as it is the only difficult token.

A natural remedy is to reduce teacher forcing via future prediction approaches, which require the model to predict tokens further ahead. This makes shortcuts less effective, as multiple lookups in the adjacency list would be needed to predict future tokens. Hence, we consider the MTP approach, which predicts the immediate future token, and the handcrafted summary method FSP-BoW, which compresses the information about all the future tokens from the path. By summarizing the entire future trajectory, FSP-BoW substantially reduces teacher forcing, encouraging the model to plan the full path instead of exploiting shortcuts.

We conduct experiments for two graphs $G(2, 6)$ and $G(2, 8)$, with all the approaches pretrained from scratch using the GPT-Mini architecture (details in Appendix C.1). At inference, we discard the auxiliary head used in future prediction, and the task is to generate the complete path given the prefix $p$. The evaluation metric checks whether the generated path exactly matches the true path. Results in Figure 3 (left) show that both NTP and MTP fail to generalize (both obtained perfect training accuracy), and the accuracy of MTP degrades further for the scenario with longer path $G(2, 8)$. Hence, predicting just the immediate future token is not enough, and FSP-BoW tackles this by efficiently compressing all the future tokens from the path, enabling it to achieve perfect accuracy in both cases. Appendix C.1 (Table 4) shows that increasing the number of auxiliary heads in MTP

can provide some improvement, but practical limits exist: even with four additional future heads MTP cannot solve the longer path graph $G(2, 8)$.

## 3.2 Adaptive future summary is important

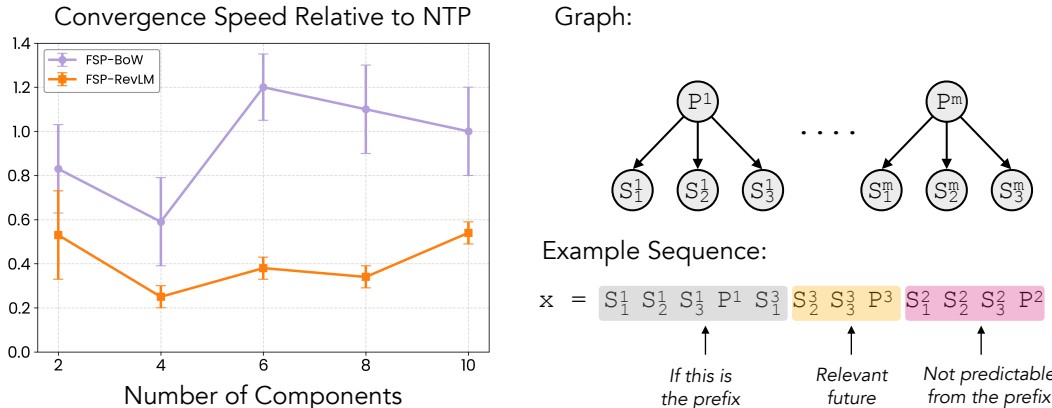

Figure 4: *Analysis on the modified sibling discovery task which requires adaptive future summaries.* **Left:** Convergence speed (mean (s.e.) over 3 random seeds) relative to NTP, where lower values imply faster convergence. FSP-RevLM converges faster than NTP while FSP-BoW improves only in the cases with few components. **Right:** Task setup illustration– given the prefix, only future tokens in the highlighted component are informative. FSP-BoW, which summarizes all future tokens, suffers from irrelevant information, whereas FSP-RevLM summarizes only the informative aspects.

Long handcrafted summaries often incorporate all future tokens, even though only a subset may provide meaningful supervision signals, while the rest can introduce noise. This motivates the need for an *adaptive* future summary. We illustrate this with modification of the sibling discovery task. Figure 4 (right) depicts the setup, the model must generate sequences made of concatenated, independent components, where each component lists its children nodes first, followed by their parent (e.g., nodes $S_1^1, S_2^1, S_3^1$ followed by parent $P^1$).

Intuitively, the causal factorization implied by the DAG (parent followed by children) implies a goal-conditioned approach: by conditioning on the parent, the model can easily capture the sibling dependencies. Under NTP, the model estimates $P_\theta(S_2^1|S_1^1)$ without the parent, making sibling relationships harder to learn and requiring more samples. Future prediction can help, when the model predicts the parent from $S_1^1$, then the representation can incorporate the parent information. This enables goal-conditioned planning, the model can predict $S_2^1$ conditioned on both $S_1^1$ and the parent, allowing sample efficient learning of sibling relationships (see Nagarajan et al. (2025) for details).

However, not all future tokens are equally informative. As Figure 4 illustrates, future tokens from a different component do not provide relevant signal for predicting $S_2^1$. Handcrafted summaries that include all future tokens may therefore be affected by the irrelevant information, whereas learned summaries remain robust, as the reverse language model learns representation that emphasize only the predictive signals needed to infer the next token.

We verify this empirically by comparing FSP-BoW and FSP-RevLM in experiments with varying number of total components. All models are pretrained from scratch with the GPT-Mini architecture (details in Appendix C.2), and the inference task requires generating a coherent sequence (siblings followed by parent for each component). At convergence, all models produce coherent sequences; to quantify the learning speedup over NTP, we report the ratio of steps to convergence relative to NTP, with lower values indicating faster learning. Since all methods eventually reach perfect consistency at convergence, time to convergence is a reliable metric for comparison. Results in figure 4 (left) shows that FSP-BoW improves over NTP only when the number of components is small, with gains disappearing for more than six components. In contrast, FSP-RevLM consistently achieves faster convergence across all component sizes, confirming the benefit of adaptive future summaries.

## 4 EXPERIMENTS

### 4.1 SETUP

We pretrain 3B- and 8B-parameter models on corpora of 250B and 1T tokens, respectively, covering diverse domains. The majority of the data comes from DCLM-like sources and GitHub repositories, supplemented by specialized material in mathematics, programming, and related areas. Models are evaluated across a diverse set of benchmarks: ARC-Easy/Challenge (Clark et al., 2018) for general reasoning, MBPP (Austin et al., 2021) and HumanEval+ (Liu et al., 2023) for code generation, and GSM8K (Cobbe et al., 2021) and Math-500 (Hendrycks et al., 2021) for mathematical reasoning. Details of the pretraining corpus and hyperparameters are provided in Appendix D.1.

We first benchmark our primary method, FSP-RevLM, against the baselines: next-token prediction (NTP), standard multi-token prediction (MTP) (Gloeckle et al., 2024), and DeepSeek-MTP (DS-MTP) (Liu et al., 2024). Further, a natural way to improve MTP for longer horizons is to add multiple auxiliary heads, each predicting a token farther into the future, but this approach quickly becomes impractical. We therefore constrain both MTP and DS-MTP to a single auxiliary head predicting the immediate future token. This design choice keeps the comparison consistent and aligned with the proposed FSP, since it uses a single auxiliary head.

Building on this unified single auxiliary head framework, we conduct an analysis of how MTP can be enhanced by predicting richer future targets instead of the immediate future token. In addition to predicting the learned future summary (FSP-RevLM), we compare handcrafted future summaries over short- and long-range windows. This includes the proposed multi-hot future summary (FSP-BoW) as a contributed baseline, and a random-token summary baseline (radomly sampling token from future), in line with prior works (Thankaraj et al., 2025; Gerontopoulos et al., 2025).

**Note regarding experiment design.** All experiments are conducted under iso-data conditions, meaning that all the methods are trained on identical datasets. For the proposed FSP-RevLM, this implies that both the forward and reverse models are trained on the same data. In line with standard practice in distillation, we do not perform iso-compute comparisons that include the teacher model's (ReverseLM) cost in the reported compute budget. In practical scenarios, the computational cost of training the teacher model is typically amortized, hence it can be treated as a one-time overhead that is excluded from comparisons of student models (Gemma et al., 2024; 2025).

**Note regarding FSP-RevLM implementation.** In our experiments with FSP-RevLM, the reverse model is the same size as the forward model (and other baselines), and it is trained for the same number of steps. As a result, FSP-RevLM roughly doubles the total compute cost compared to standard NTP training. While FSP-RevLM increases training FLOPs, we believe this tradeoff is reasonable in today's compute-rich, data-limited scaling regime. The field has effectively hit the data wall, whereas available compute continues to grow. Progress increasingly depends on using this growing compute-per-token budget to extract more value from fixed datasets. In this context, methods that deliver measurable gains without requiring additional data, even at higher compute cost, are valuable.

### 4.2 RESULTS

At the 8B scale (Table 1), future-summary supervision via the ReverseLM (FSP-RevLM) consistently improves the performance across different evaluation tasks. On ARC-Easy, FSP-RevLM (76.6%) provides significant improvement over NTP (71.8%) and MTP (73.6%), and it also leads on ARC-Challenge and MATH. For code generation, it achieves the highest score on MBPP and ties with MTP on HumanEval+, showing that the benefits of predicting future summaries generalize across both reasoning and program synthesis tasks. GSM8K is the one task where NTP (71.6%) holds a lead, though FSP-RevLM (70.5%) still narrows the gap relative to MTP (67.8%).

At the 3B scale (Table 2), DeepSeek-MTP is a strong baseline and obtains better performance than FSP-RevLM on most tasks, except math reasoning. More importantly, FSP-RevLM exhibits larger relative improvements as scale increases from 3B to 8B parameters, and becomes more favourable than DS-MTP. Further, note that even at the 3B scale, FSP-RevLM still beats MTP on ARC and math reasoning tasks, and performs comparably on MBPP, suggesting that even at smaller scales,

| Task | NTP | MTP | DS-MTP | FSP-RevLM |
|---|---|---|---|---|
| ARC-Easy | 0.718 (0.000) | 0.736 (0.000) | 0.617 (0.003) | **0.766** (**0.000**) |
| ARC-Challenge | 0.531 (0.000) | 0.552 (0.000) | 0.426 (0.002) | **0.559 (0.000)** |
| GSM8K | **0.716** (**0.003**) | 0.678 (0.007) | 0.704 (0.003) | 0.705 (0.004) |
| MATH | 0.342 (0.008) | 0.309 (0.006) | 0.335 (0.014) | **0.351 (0.017)** |
| MBPP | 0.657 (0.004) | 0.672 (0.008) | 0.678 (0.006) | **0.683 (0.006)** |
| HumanEval+ | 0.478 (0.019) | **0.541 (0.011)** | 0.526 (0.013) | **0.541 (0.009)** |

Table 1: **Pretraining at 8B scale.** We benchmark the proposed FSP-RevLM approach against NTP, MTP, and DS-MTP. Results (mean $\pm$ s.e. over 3 seeds) report pass@16 for code/math tasks and accuracy for ARC. FSP-RevLM achieves the strongest overall performance, with large gains over the baselines on ARC tasks and MATH, and competitive results with (DS) MTP on code benchmarks.

| Task | NTP | MTP | DS-MTP | FSP-RevLM |
|---|---|---|---|---|
| ARC-Easy | 0.263 (0.002) | 0.272 (0.005) | **0.293 (0.000)** | 0.277 (0.000) |
| ARC-Challenge | 0.263 (0.001) | 0.245 (0.002) | **0.274 (0.008)** | 0.255 (0.000) |
| GSM8K | 0.410 (0.003) | 0.411 (0.001) | 0.417 (0.003) | **0.436** (**0.003**) |
| MATH | **0.213 (0.004)** | 0.196 (0.009) | 0.201 (0.004) | **0.212 (0.002)** |
| MBPP | 0.521 (0.007) | 0.526 (0.004) | **0.537 (0.007)** | 0.524 (0.001) |
| HumanEval+ | 0.301 (0.009) | 0.321 (0.015) | **0.348** (**0.022**) | 0.305 (0.006) |

Table 2: **Pretraining at 3B scale.** We benchmark the proposed FSP-RevLM approach against NTP, MTP, and DS-MTP. Results (mean $\pm$ s.e. over 3 seeds) report pass@16 for code/math tasks and accuracy for ARC. At this smaller scale, DS-MTP is a strong overall baseline, but FSP-RevLM outperforms it on math reasoning tasks, and also provides substantial gains over MTP on ARC and math reasoning tasks. Further, as we scale the approaches to 8B parameters, FSP-RevLM scales more favorably than DS-MTP, overtaking it on most tasks.

learning to predict future summaries may provide more effective auxiliary signal than immediate future token prediction with MTP.

### 4.3 ANALYZING MTP WITH DIFFERENT FUTURE SUMMARIES

Table 3 presents our analysis of different future-summary strategies as auxiliary head targets at 8B scale. We focus on the standard MTP architecture, without comparing to DS-MTP as it modifies the input to auxiliary head, to isolate the effect of different future targets on the auxiliary head.

Our results show that random-token handcrafted future summaries perform worse than standard MTP with immediate future token prediction, and performance further degrades as the future window ($\tau$) increases. In contrast, the proposed multi-hot or bag-of-words handcrafted future summaries yield meaningful improvements over MTP, especially on math reasoning tasks, with both shorter ($\tau = 12$) and longer ($\tau = 100$) future windows. For example, FSP-BoW with $\tau = 12$ achieves 33.1% on MATH (+2.2 points) and 69.9% on GSM8K (+2.1 points), while even longer windows ($\tau = 100$) further amplifies the performance on GSM8K (71.4%, +3.6 points).

Finally, our learned future summaries (FSP-RevLM) outperform MTP across all evaluation tasks, with especially pronounced gains on math reasoning: 35.1% on MATH (+4.2 points) and 70.5% on GSM8K (+3.5 points). Further analysis (Figure 5) for these math reasoning tasks shows that learned summaries promote greater output diversity across different pass@k settings, compared to vanilla MTP with immediate future token prediction.

In Appendix D.2 (Table 6), we replicate these findings at 3B scale, where both handcrafted and learned future summaries improve over vanilla MTP, again most prominently on math reasoning tasks. Additional ablations explore the effects of omitting reweighting in FSP-BoW and predicting learned summaries from deeper layers of the ReverseLM (FSP-RevLM).

| Method | MBPP | GSM8K | MATH | HumanEval+ | ARC-Challenge | ARC-Easy |
|---|---|---|---|---|---|---|
| MTP | 0.672 (0.008) | 0.678 (0.007) | 0.309 (0.006) | **0.541 (0.011)** | 0.552 (0.000) | 0.736 (0.000) |
| MTP-Skip $\tau$:4 | 0.658 (0.005) | 0.639 (0.004) | 0.277 (0.020) | 0.508 (0.009) | 0.494 (0.003) | 0.722 (0.000) |
| MTP-Skip $\tau$:12 | 0.623 (0.002) | 0.621 (0.005) | 0.287 (0.018) | 0.486 (0.010) | 0.512 (0.000) | 0.710 (0.003) |
| MTP-Skip $\tau$:32 | 0.611 (0.008) | 0.598 (0.007) | 0.271 (0.005) | 0.459 (0.007) | 0.379 (0.000) | 0.564 (0.000) |
| FSP-BoW $\tau$:12 | 0.669 (0.005) | 0.699 (0.006) | 0.331 (0.016) | 0.508 (0.005) | **0.562 (0.000)** | 0.737 (0.000) |
| FSP-BoW $\tau$:100 | 0.671 (0.002) | **0.714 (0.009)** | 0.331 (0.007) | 0.500 (0.019) | 0.459 (0.000) | 0.662 (0.000) |
| FSP-RevLM | **0.683 (0.006)** | 0.705 (0.004) | **0.351 (0.017)** | 0.541 (0.009) | 0.559 (0.000) | **0.766 (0.000)** |

Table 3: **Analysis of future-summary strategies at 8B scale.** We evaluate the effect of different future-summary prediction approaches against vanilla MTP. Results (mean $\pm$ s.e. over 3 seeds) report pass@16 for code/math tasks and accuracy for ARC. Handcrafted multi-hot summaries (FSP-BoW) improve over standard MTP, especially on math reasoning (e.g., GSM8K and MATH), while learned summaries (FSP-RevLM) provide the largest gains across math reasoning and ARC tasks.

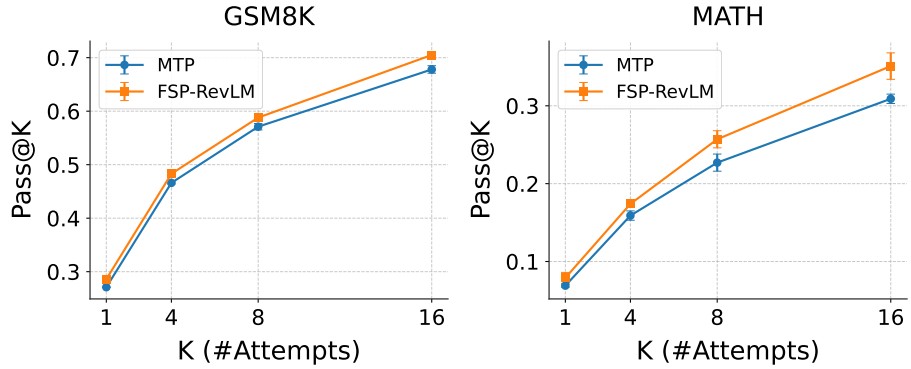

Figure 5: **Enhanced diversity through learned future summaries.** We compare standard MTP, which predicts the immediate future token, with FSP-RevLM, which enriches the auxiliary head target using learned future summaries. FSP-RevLM substantially increases output diversity compared to MTP on GSM8K and MATH benchmarks.

## 5   RELATED WORK

**Beyond Multi-token Prediction.**   Multi-token Prediction (MTP) extends Next-token Prediction (NTP) by introducing auxiliary heads to predict future tokens (Gloeckle et al., 2024), thereby providing a richer learning signal. Variants such as DeepSeek-MTP (Liu et al., 2024), joint-token prediction (Ahn et al., 2025), and next-latent prediction (Teoh et al., 2025) differ in how they structure these auxiliary predictions, but they retain a common limitation: scaling auxiliary heads to capture long-range dependencies becomes impractical. More recent approaches attempt to improve efficiency by predicting randomly selected or non-sequential future tokens (Gerontopoulos et al., 2025; Thankaraj et al., 2025; Liu et al., 2025). However, heuristically sampling future tokens risks missing the most informative long-range signals. Our approach (FSP-RevLM) addresses this by *predicting a learned summary of the future*, such that it can extract meaningful long-range information.

Realted to the proposed hand-crafted summary approach (FSP-BoW), (Yin et al., 2024) and Frydenlund (2025) incorporate objectives that predict an unordered set-level summary of future tokens, conceptually similar to our FSP-BoW and its binary cross-entropy formulation. However, our formulation is more general as we allow re-weighting schemes such as tf-idf scores within the BCE objective and is implemented via auxiliary heads to be consistent with the multi-token prediction architecture (Gloeckle et al., 2024). More importantly, our contribution is not limited to a specific BoW variant: we introduce a unified framework for future summary prediction that clarifies the trade-offs between different forms of future supervision. For instance, a BoW summary directly resolves the path–star graph failure by aggregating the long-range information, yet it remains suboptimal when only part of the future is relevant, as observed in our sibling discovery experiments.

These insights motivate moving beyond hand-crafted summaries toward a learned future summary via reverse LM, which adaptively captures the most informative aspects of the future.

Finally, regarding SemFormer (Yin et al., 2024), beyond its BoW-style supervision, its central idea is to introduce special planning tokens that are trained to predict a latent embedding of the future sequence, encouraging planning. While conceptually related, it applies future supervision only at designated planning tokens and relies on auto-encoding objective to learn future embeddings. In contrast, our approach requires no special planning tokens, enforces future-summary alignment at every position in the sequence through auxiliary heads, and, learns future embeddings via the ReverseLM, yielding a fundamentally different supervision signal.

**Leveraging the right-to-left signal.** Leveraging reverse, or "right-to-left", order during training has shown empirical benefits across multiple learning paradigms. The Belief State Transformer (BST) (Hu et al., 2024) employs dual forward and backward encoders to predict both the next token after a prefix and the previous token before a suffix. This bidirectional training encourages the model to form a compact belief state, though it does not explicitly address teacher forcing. In contrast, our FSP-RevLM incorporates a reverse model with a different objective: mitigating dependence on teacher forcing. While trained on standard left-to-right sequences, FSP-RevLM aligns the forward model's embeddings with those of a reverse model, effectively distilling the reverse order into the forward language model.

The most closely related work aimed at reducing teacher forcing is Twin Networks (Serdyuk et al., 2017), which trains a reverse RNN and matches the forward hidden states to those of the reverse model to encourage long-range future dependence. While similar at a high level, our contributions go beyond the specific FSP-RevLM mechanism: we present a broader perspective in which future summary prediction serves as a framework for understanding and designing pretraining objectives, together with evidence showing when simpler approaches fail and why a learned summary coupled with a reverse LM becomes necessary. Moreover, scaling this idea to Transformers is non-trivial. Just as TwinNet anticipated aspects of our reverse component, earlier work also explored multiple future tokens or parallel token blocks prediction from a given prefix (Tschannen et al., 2023; Monea et al., 2023) well before Gloeckle et al. (2024), though without establishing MTP as a broadly effective objective for large-scale Transformers. Gloeckle et al. (2024) deserve credit for identifying a formulation that works in modern LLM training, and our results extend this line by showing that both MTP and future-summary prediction can be cleanly integrated into Transformer pretraining and scaled to yield robust gains.

Another related approach, Meet-in-the-Middle (MiM) (Nguyen et al., 2023), jointly trains forward and backward models with shared parameters and employs an agreement regularizer to align their output distributions. Our method differs in two key aspects: (1) we perform distillation with the reverse model rather than parameter sharing, and (2) we avoid the somewhat impractical assumption required by MiM that the forward and reverse output distributions must match exactly.

## 6 CONCLUSION

In this work, we highlighted a key limitation of existing multi-token prediction methods: the difficulty of scaling auxiliary heads for long-horizon future prediction. Towards this, we proposed *Future Summary Prediction* (FSP), a novel pretraining framework that shifts the auxiliary objective from predicting specific future tokens to predicting a (learned) summary of the future. Experiments on 8B models, using both hand-crafted and learned summaries, demonstrate that FSP delivers a stronger, more robust training signal, improving over NTP and MTP on challenging reasoning and coding tasks. Our findings indicate that focusing on abstract, predictable aspects of the future is a promising strategy for designing more efficient and effective pretraining objectives for next-generation large language models. Future work can focus on designing computationally efficient approaches for learning future summaries (embeddings), potentially without training a reverse model.

ACKNOWLEDGMENTS

The authors thank Sachin Mehta for suggestions on the real-world pretraining experimental setup, and thank Vaishnavh Nagarajan and Nanda H Krishna for their detailed feedback on the draft. The authors also gratefully acknowledge helpful discussions with Aniket Didolkar, Andrei Nicolicioiu, Mathurin Videau, Sarthak Mittal, Sharut Gupta, Vineet Jain, and Moksh Jain. This research was fully funded by Meta through Meta's AI Mentorship (AIM) program with Mila. Ioannis Mitliagkas in his role as Divyat Mahajan's academic advisor, acknowledges support by an NSERC Discovery grant (RGPIN-2019-06512), and a Canada CIFAR AI chair. Divyat Mahajan acknowledges support via FRQNT doctoral scholarship (`https://doi.org/10.69777/354785`) for his graduate studies.

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

## A    EXTENDED RELATED WORKS

**Pitfalls of Next-token Prediction.**    While next-token prediction (NTP) is the standard loss for modern LLMs, its shortcomings are increasingly evident. A core issue is the train–test mismatch: training conditions on ground-truth tokens (teacher forcing), while inference conditions on the model's own generations (autoregression), leading to exposure bias and compounding errors over long horizons (Bengio et al., 2015; Arora et al., 2022). Beyond the auto-regressive error accumulation, teacher forcing may encourage learning spurious correlations or "shortcuts" rather than the relevant learning signal. Bachmann & Nagarajan (2024) describe this as the "Clever Hans cheat," where the model exploits easy local cues in the prefix that correlate with the target, thereby absorbing gradient signal that would otherwise teach genuinely lookahead-dependent structure in tasks such as their path-star graph problem. Nagarajan et al. (2025) further argue that NTP is *myopic and data-inefficient* on tasks requiring an implicit long-range "leap of thought" (e.g., Sibling Discovery) such as their sibling discovery task. Our approach tackles these failures by providing a more robust, long-range training signal that discourages such shortcuts.

**Beyond Multi-token Prediction.**    Multi-token Prediction (MTP) addresses limitations of Next Token Prediction (NTP) by using auxiliary heads to predict future tokens (Gloeckle et al., 2024). Within this MTP family, DeepSeek MTP (Liu et al., 2024) uses a recursive auxiliary design: when predicting a future token, each auxiliary head conditions on the backbone state and the intermediate representations produced by earlier auxiliary heads. Closely related, Joint Token Prediction (Ahn et al., 2025) keeps a similar slight-teacher-forcing flavor but predicts all future tokens in parallel, while Next Latent Prediction (Teoh et al., 2025) replaces discrete future-token targets with a representation-space alignment objective that matches the current hidden state to future hidden states. But the key issue with all these approaches is that scaling the auxiliary heads for long-range dependencies is impractical. Recent work aims to improve efficiency and capture longer dependencies by radomly predicting information from the future sequence. Gerontopoulos et al. (2025) introduce register tokens, predicting tokens $k$ steps ahead without architectural changes. Thankaraj et al. (2025) insert lookahead tokens containing future subsequences, while Liu et al. (2025) use a leap-based strategy to predict non-sequential future tokens. However, heuristically sampling random future tokens still poses the risk of missing informative long-range signals. Our FSP-RevLM addresses this by *predicting a learned summary of the future*, such that it can extract meaningful long-range information.

A relevant predecessor from the RNN literature that in principle avoids scaling auxiliary heads is ProphetNet (Qi et al., 2020), which uses a shared self-attention mechanism together with relative positional encodings to predict future n-grams without introducing separate heads. While this may seem related to our bag-of-words summary (FSP-BoW), ProphetNet still supervises each future token individually and preserves positional order. In contrast, FSP-BoW operates on a summary vector that discards positional information, and applies a binary cross-entropy objective over an unordered future-token summary, yielding a fundamentally different training signal.

Further, (Yin et al., 2024) and Frydenlund (2025) incorporate objectives that predict an unordered set-level summary of future tokens, conceptually similar to our FSP-BoW and its binary cross-entropy formulation. However, our formulation is more general as we allow re-weighting schemes such as tf-idf scores within the BCE objective and is implemented via auxiliary heads to be consistent with the multi-token prediction architecture (Gloeckle et al., 2024), rather than as a standalone modification. More importantly, our contribution is not limited to a specific BoW variant: we introduce a unified framework for future summary prediction that clarifies the trade-offs between different forms of future supervision. For instance, a BoW summary directly resolves the path–star graph failure by aggregating the long-range information, yet it remains suboptimal when only part of the future is relevant, as observed in our Sibling Discovery experiments. These insights motivate moving beyond hand-crafted summaries toward a learned future summary via reverse LM, which adaptively captures the most informative aspects of the future.

Finally, regarding SemFormer (Yin et al., 2024), beyond its BoW-style supervision, its central idea is to introduce special planning tokens that are trained to predict a latent embedding of the future sequence, encouraging planning. While this shares the high-level goal of learning a future representation as in our approach FSP-RevLM, there are several differences. SemFormer applies the embedding-matching loss only at designated planning tokens (typically once per sequence), limiting

where future information is explicitly enforced. In contrast, our approach introduces no planning tokens and enforces the future embedding matching loss at every position in the sequence via the auxiliary prediction head, which enables us to employ an architecture that is fully aligned with the multi-token prediction approach (Gloeckle et al., 2024). Moreover, SemFormer learns future embeddings via the auto-encoding objective (left-to-right order of future sequence), whereas our FSP-RevLM computes the summary using a reverse LM (right-to-left order of future sequence), resulting in a fundamentally different supervision signal.

**Leveraging the right-to-left signal.** Leveraging reverse, or "right-to-left", order during training has shown empirical benefits across multiple learning paradigms. Reverse Training (Golovneva et al., 2024) augments the dataset with reversed sequences to teach bidirectional dependencies and mitigate the Reversal Curse (Berglund et al., 2023). The Belief State Transformer (BST) (Hu et al., 2024) employs dual forward and backward encoders to predict both the next token after a prefix and the previous token before a suffix. This bidirectional training encourages the model to form a compact belief state, though it does not explicitly address teacher forcing. In contrast, our FSP-RevLM incorporates a reverse model with a different objective: mitigating dependence on teacher forcing. While trained on standard left-to-right sequences, FSP-RevLM aligns the forward model's embeddings with those of a reverse model, effectively distilling the reverse order into the forward language model. Another related approach, Meet-in-the-Middle (MiM) (Nguyen et al., 2023), jointly trains forward and backward models with shared parameters and employs an agreement regularizer to align their output distributions. Our method differs in two key aspects: (1) we perform distillation with the reverse model rather than parameter sharing, and (2) we avoid the somewhat impractical assumption required by MiM that the forward and reverse output distributions must match exactly.

The most closely related work aimed at reducing teacher forcing is Twin Networks (Serdyuk et al., 2017), which trains a reverse RNN and matches the forward hidden states to those of the reverse model to encourage long-range future dependence. While similar at a high level, our contributions go beyond the specific FSP-RevLM mechanism: we present a broader perspective in which future summary prediction serves as a framework for understanding and designing pretraining objectives, together with evidence showing when simpler approaches fail and why a learned summary coupled with a reverse LM becomes necessary. Moreover, scaling this idea to Transformers is non-trivial. Just as TwinNet anticipated aspects of our reverse component, earlier work also explored multiple future tokens or parallel token blocks prediction from a given prefix (Tschannen et al., 2023; Monea et al., 2023) well before Gloeckle et al. (2024), though without establishing MTP as a broadly effective objective for large-scale Transformers. Gloeckle et al. (2024) deserve credit for identifying a formulation that works in modern LLM training, and our results extend this line by showing that both MTP and future-summary prediction can be cleanly integrated into Transformer pretraining and scaled to yield robust gains.

Another somewhat related line is z-forcing (Goyal et al., 2017), which uses a reverse RNN to infer latent variables that are then injected into the forward RNN to compute its hidden states. In contrast, our reverse LM is not performing latent-variable inference; it is used solely to generate targets for an auxiliary prediction head. Furthermore, unlike Z-forcing, our approach does not require the reverse model at inference time—the reverse LM is purely a training-time component, whereas Z-forcing depends on the reverse RNN during inference as well.

## B    DETAILS ON MULTI-TOKEN PREDICTION VARIANTS

In Section 2, we discussed MTP and the proposed future summary prediction to address a major limitation in MTP. We summarize other variants proposed in the literature and their limitations here.

**Incorporating slight teacher forcing in MTP.**    Note that in the standard MTP we did not allow for any teacher forcing from the "auxiliary" tokens, i.e, the auxiliary head predicted future tokens $x_{t+k}$ given only the prefix $x_{\leq t}$. This was modified by a popular variant proposed by DeepSeek (Liu et al., 2024), where we condition the auxiliary head $f'_{h_k}$ with full prefix $x_{\leq t+k-1}$ to predict the future tokens $x_{t+k}$, but with reduced teacher forcing from the "auxiliary" tokens $\{x_{t+1}, x_{t+2}, \cdots, x_{\leq t+k-1}\}$ as well.

$$L_{\text{DS-MTP}}(X, P_\theta) = -\mathbb{E}_{x \sim \mathbb{P}_X}\left[\sum_{t=1}^{T-1}\sum_{k=1}^{\tau}\mathbf{1}[t+k \leq T]\log P_\theta(x_{t+k} \,|\, x_{\leq t+k-1})\right]. \qquad (14)$$

The parameterization is:

$$\begin{aligned} P_\theta(x_{t+1} \,|\, x_{\leq t}) &= \text{softmax}\Big(f_u \circ f_h \circ f_s\big(x_{\leq t}\big)\Big), \\ P_\theta(x_{t+k} \,|\, x_{\leq t+k-1}) &= \text{softmax}\Big(f_u \circ f'_{h_k}\Big(f_s\big(x_{\leq t}\big), x_{t+1}, \ldots, x_{t+k-1}\Big)\Big), \quad \forall k > 1. \end{aligned} \qquad (15)$$

Here, the auxiliary (future) tokens $(x_{t+1}, \ldots, x_{t+k-1})$ are injected directly into the auxiliary heads, bypassing the backbone, which reduces teacher forcing compared to standard NTP.

A notable architectural feature of DS-MTP is its recursive nature: the auxiliary head $f'_{h_t}$ takes as input not only the backbone representation $f_s(x_{\leq t})$ but also the representations produced by the previous auxiliary heads $\{f'_{h_1}, \cdots, f'_{h_{k-1}}\}$ when predicting the future token $x_k$, creating a chain of dependent predictions across depths.

Joint-token prediction (Ahn et al., 2025) shares a similar flavor of slight teacher forcing but computes all future token predictions in parallel rather than recursively (same parametrization as 15), which the authors argue encourages the model to maintain a belief state over future tokens jointly. Another related variant, Next Latent Prediction (Teoh et al., 2025), also incorporates slight teacher forcing in a similar spirit, but differs in its objective: rather than predicting discrete future tokens, it supervises the model's current hidden state to match the hidden states the model would produce at future positions, using a self-supervised alignment loss operating purely in representation space.

However, all these approach suffers from the same short-horizon limitation as MTP: predicting multiple future steps would require adding separate auxiliary heads for each token, which quickly becomes impractical and limits scalability to long future sequences.

**Randomly sampled future information.**    In this work, we emphasized that standard MTP and DS-MTP often fail to capture the most informative future tokens, as both primarily focus on predicting the immediate next tokens via the auxiliary head. The TRELAWNEY approach of Thankaraj et al. (2025) adopts a complementary strategy: rather than altering the model architecture, it augments the training data by inserting a window of future tokens, bounded by special markers. This encourages the model to predict a randomly sampled block of future tokens, thereby reducing reliance on teacher forcing. Moreover, TRELAWNEY allows random subsequences to be sampled from the future, with the goal of exposing the model to a richer and more informative signal than what is available in the immediate next few steps. In a similar spirit, MuToR (Gerontopoulos et al., 2025) interleaves learnable register tokens into the input sequence, each tasked with predicting a future token at a randomly sampled offset, and L-MTP (Liu et al., 2025) introduces a leap-based mechanism that skips over adjacent tokens and instead predicts non-sequential future tokens in a single forward pass, aiming to capture longer-range dependencies while also enabling inference acceleration.

However, all these approaches based on heuristic random sampling can still fail to capture critical information from tokens far in the future, limiting its robustness for long-horizon planning tasks.

## C    SYNTHETIC EXPERIMENTS

### C.1    PATH-STAR GRAPH

**Experiment Setup.**    We adopt the dataset generation procedure from the official code repository[1] of Bachmann & Nagarajan (2024). Specifically, we construct a set of 50 distinct nodes and generate multiple instances of path-star graph sequences by randomly sampling nodes from this set. The training set consists of $n_{\text{train}} = 200,000$ sequences, and evaluation is performed on $n_{\text{test}} = 20,000$ sequences.

For modeling, we employ the GPT-Mini architecture with the following configuration:

- Total Layers: 12
- Embedding Dimension: 384
- Total Attention Heads: 6
- MLP expansion factor: 4

We summarize the other relevant hyperparameters below.

- Learning Rate: $3e-4$
- Batch Size: 256
- Weight Decay: $1e-2$
- Total Epochs: 500
- Gradient Norm Clipping: 1.0

**Experiment with multiple auxiliary heads in MTP.**    A naive approach to incorporate longer-term future dependencies in MTP is by increasing the number of auxiliary heads. Table 4 reports the results: adding more auxiliary heads improves performance on $G(2,6)$, due to reduced teacher forcing. However, simply scaling the number of auxiliary heads is not a practical solution for modeling long-range dependencies, as shown by the poor performance on the longer path graph $G(2,8)$.

| Method | G(2,6) | G(2,8) |
|---|---|---|
| NTP | 0.45 (0.01) | 0.45 (0.03) |
| MTP | 0.66 (0.17) | 0.48 (0.03) |
| MTP (aux heads: 2) | 0.89 (0.09) | 0.46 (0.01) |
| MTP (aux heads: 3) | 0.98 (0.01) | 0.47 (0.06) |
| MTP (aux heads: 4) | 0.97 (0.01) | 0.48 (0.12) |
| FSP-BoW | 1.00 (0.00) | 1.00 (0.00) |

Table 4: *Analyzing the performance of MTP on path-star graphs with an increasing number of auxiliary heads.* Augmenting MTP with additional auxiliary heads improves performance on $G(2,6)$, but practical limits exist: even with 4 aux heads, MTP fails to solve the longer path graph $G(2,8)$.

### C.2    SIBLING DISCOVERY

**Experiment Setup.**    We build on the dataset generation procedure from the official code repository[2] of Nagarajan et al. (2025). While the original sibling discovery task presents sequences from a single component, we modify the setup by concatenating sequences from multiple components. Each component $i$ is defined by a unique parent $P^i$ and three child nodes $S_1^i, S_2^i, S_3^i$. The support of each conditional distribution $P(S^i \mid P^i)$ is disjoint across children—i.e., each child $S^i$ can take $N$ possible values, and these sets of values are non-overlapping across children. As a result, with $K$ components, the total number of distinct child value combinations is $N^{3K}$. In our experiments, we vary the number of components $K \in 2, 4, 6, 8, 10$ while fixing $N = 100$. For training, we randomly

---

[1]`https://github.com/gregorbachmann/Next-Token-Failures/`
[2]`https://github.com/chenwu98/algorithmic-creativity`

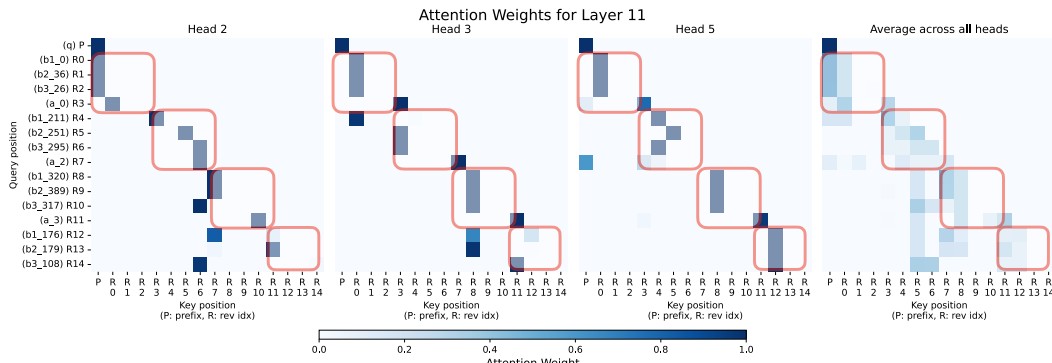

Figure 6: Each subplot corresponds to a different attention head. Red squares highlight queries belonging to a particular component, showing that they attend predominantly to keys within the same component, forming a subgroup-wise pattern. While the average across heads shows some cross-component diffusion, several heads exhibit clean intra-component attention, indicating that the reverse model preserves the structural organization of the future input rather than mixing information across unrelated parts of the sequence.

sample child values for each component to construct $n_{\text{train}} = 20{,}000$ sequences, and evaluate on $n_{\text{test}} = 3{,}000$ sequences.

We use the GPT-Mini architecture with the following specifications.

- Total Layers: 12
- Embedding Dimension: 384
- Total Attention Heads: 6
- MLP expansion factor: 4

We summarize the other relevant hyperparameters below.

- Learning Rate: $3e-4$
- Batch Size: 256
- Weight Decay: $1e-2$
- Total Epochs: 150

**Additional Experiment: Interpretability of attention weights from RevLM.** Interpreting the learned future summaries is crucial for understanding how RevLM captures information from future context. For the sibling-discovery experiment (Section 3.2), we conducted a preliminary analysis using attention weights from the final layer, which directly contribute to the future-summary representation. We observe that queries associated with a particular component in the sequence attend predominantly to keys from the same component, indicating limited cross-component interaction. While the average across heads shows some diffusion of attention beyond component boundaries, three individual heads exhibit a clear subgroup-wise structure without such leakage. This suggests that the learned future summaries preserve the informative structure of the future input by prioritizing intra-component tokens. Prior work shows that hidden states of LLMs naturally encode detailed information about surrounding tokens (see Section 5 of Physics of LLMs (Allen-Zhu & Li, 2023)), supporting the intuition that reverse-direction representations can capture meaningful long-range patterns. Extending this analysis with additional interpretability tools, such as linear probing and activation patching, remains a promising direction for future work.

# D  REAL WORLD PRETRAINING

## D.1  EXPERIMENTS SETUP

**Pretraining dataset composition.**  Our pretraining corpus is constructed to cover a wide range of domains, aiming to equip models with both broad knowledge and strong reasoning ability. The bulk of the data is drawn from a DCLM-style mixture (Li et al., 2025) and large-scale code sources such as GitHub (neogithub, 2022). To strengthen performance on more specialized skills, we further supplement with mathematics and scientific data, including the DeepMind Mathematics dataset (Saxton et al., 2019), Proof Pile 2 (ArXiv, OpenWebMath, Algebraic Stack) (Azerbayev et al., 2023), and Stack Exchange from the Pile (Gao et al., 2020). Additional curated resources include FineWeb-Edu (Lozhkov et al., 2024), the Natural Reasoning Dataset (Yuan et al., 2025), and AQuA (Ling et al., 2017). Together, this mixture balances large-scale general text with carefully chosen reasoning-focused datasets.

**Model Architecture.**  Both the 3B and 8B models follow the LLaMA 3 architecture. The 8B configuration aligns with the official LLaMA setup, while the 3B model is a scaled-down variant that preserves architectural ratios.

| Parameter | 3B Model | 8B Model |
|---|---|---|
| Hidden dimension ($d$) | 2432 | 4096 |
| Layers ($L$) | 26 | 32 |
| Attention heads | 19 | 32 |
| KV heads | 19 | 8 (GQA) |
| FFN expansion | $4\times$ | $4\times$ |
| FFN multiplier | 1.3 | 1.3 |
| Sequence length | 2048 | 8192 |
| RoPE $\theta$ | 10,000 | 500,000 |

Table 5: Architectural configurations for the 3B and 8B models.

- **Grouped Query Attention (8B only).** The 8B model uses Grouped Query Attention (32 query heads, 8 KV heads; 4:1 ratio), reducing KV-cache memory by $4\times$ during inference and enabling longer context lengths without proportional memory growth.

- **Extended RoPE.** The 8B model uses `rope_theta`=500,000 to support 8K context length, following LLaMA 3's long-context scaling strategy. The 3B model uses the standard $\theta = 10,000$.

- **SwiGLU Feed-Forward Networks.** Feed-forward blocks use SwiGLU activations with a $4\times$ expansion and a 1.3 multiplier to preserve effective hidden dimensionality under gating.

- **RMSNorm (Pre-Norm).** We adopt pre-norm RMSNorm ($\epsilon = 10^{-5}$) for improved stability and computational efficiency compared to LayerNorm.

- **Depth-Scaled Initialization.** Parameters are initialized using depth-aware scaling proportional to $1/\sqrt{2L}$, improving stability for deep networks.

- **Attention Implementation.** Attention is implemented using PyTorch Scaled Dot-Product Attention (SDPA), which dispatches to optimized kernels (e.g., FlashAttention-2) depending on hardware.

- **Mixed Precision and Parallelism.** Training uses BF16 precision for numerical stability and memory efficiency. We employ Fully Sharded Data Parallel (FSDP), which shards parameters, gradients, and optimizer states across GPUs to enable large-scale training.

**Hyperparameters.** All models were trained on NVIDIA H200 GPUs using fully sharded data parallelism (FSDP) for memory-efficient scaling. We use a cosine learning rate schedule with linear warmup at the start of training, followed by smooth cosine decay to zero. The initial learning rate is set to $3 \times 10^{-3}$ for 3B models and $3 \times 10^{-4}$ for 8B models, reflecting standard scaling practices for larger model sizes. Unless otherwise specified, models are trained with BF16 mixed precision, gradient clipping for stability, and AdamW optimization. Main hyperparameters are listed below.

- **3B Models**
    - Batch Size (per GPU): 16
    - Total GPUs: 128
    - Sequence Length: 2048
    - Total Training Steps: 60k
- **8B Models**
    - Batch Size (per GPU): 2
    - Total GPUs: 256
    - Sequence Length: 8192
    - Total Training Steps: 240k

Note that these choices lead to total tokens seen during training as 250B tokens for the 3B parameter model, and 1T tokens for the 8B parameter model.

**Evaluation Metrics.** We evaluate all tasks using the pass@k metric. For multiple-choice benchmarks such as ARC (Easy and Challenge), we report pass@1, which is equivalent to standard accuracy. For mathematical reasoning and code generation benchmarks, we report pass@k with $k > 1$, measuring whether at least one of $k$ independently sampled generations solves the problem correctly. All results are reported as the mean and standard error over 3 independent random seeds.

For pass@k evaluation, we generate samples across a range of decoding temperatures $(0, 0.1, 0.2, \dots, 0.9, 1.0)$ while keeping other decoding hyperparameters fixed. For each method and task, we report the performance corresponding to the temperature that yields the highest pass@k. This ensures a fair comparison, as different training objectives may prefer different temperatures.

Importantly, auxiliary prediction heads are used only during training to shape representation learning. At test time, all models are evaluated exclusively using their standard next-token prediction heads, ensuring that improvements stem from improved pretraining dynamics rather than additional inference-time components.

## D.2 ADDITIONAL RESULTS

### D.2.1 ANALYSIS OF FUTURE SUMMARIES (3B).

Table 6 presents our analysis of different future-summary strategies as auxiliary head targets at the 3B scale. Similar to the 8B case, we focus on the standard MTP architecture, without comparing to DS-MTP since it modifies the input to the auxiliary head, allowing us to isolate the effect of different future targets. Our findings are similar: random-skip handcrafted summaries underperform relative to MTP, while the proposed multi-hot/bag-of-words approaches perform well. In our experiments, we observed consistent improvement over MTP with the proposed handcrafted summaries (FSP-BoW) on both Math and GSM8k tasks, across the 3B and 8B model scales. For the 3B ablation, we tested removing tf-idf reweighting as well and found that, in most cases, it did not provide benefits, which led us to keep the tf-idf reweighting for our 8B experiments. The only exceptions where no tf-idf reweighting performed better were GSM8k at $\tau = 12$ and HumanEval+ at $\tau = 10$. Additionally, we experimented with using deeper layers at the 3B scale and found that the last layer performed better on ARC Challenge and ARC Easy, though it was slightly worse on MBPP and Math. Based on these findings, we chose to use the last layer for our 8B-scale experiments.

| Method | MBPP | GSM8K | MATH | HumanEval+ | ARC-Challenge | ARC-Easy |
|---|---|---|---|---|---|---|
| MTP | 0.526 (0.004) | 0.411 (0.001) | 0.196 (0.009) | **0.321** (0.015) | 0.245 (0.002) | 0.272 (0.005) |
| MTP-Skip $\tau$:4 | 0.519 (0.004) | 0.368 (0.007) | 0.181 (0.005) | 0.309 (0.004) | 0.241 (0.000) | 0.282 (0.000) |
| MTP-Skip $\tau$:12 | 0.485 (0.011) | 0.354 (0.005) | 0.177 (0.006) | 0.287 (0.009) | 0.264 (0.008) | 0.262 (0.000) |
| MTP-Skip $\tau$: 32 | 0.467 (0.007) | 0.354 (0.003) | 0.189 (0.004) | 0.278 (0.017) | 0.243 (0.000) | 0.240 (0.003) |
| FSP-BoW $\tau$:12, no tf-idf | 0.518 (0.006) | 0.431 (0.004) | 0.201 (0.009) | 0.301 (0.009) | 0.250 (0.000) | 0.251 (0.000) |
| FSP-BoW $\tau$:12 | 0.521 (0.005) | 0.419 (0.003) | 0.204 (0.002) | 0.305 (0.012) | 0.254 (0.000) | 0.262 (0.004) |
| FSP-BoW $\tau$:100, no tf-idf | 0.512 (0.007) | 0.416 (0.005) | 0.203 (0.008) | 0.309 (0.004) | 0.228 (0.003) | 0.238 (0.000) |
| FSP-BoW $\tau$:100 | 0.524 (0.004) | 0.417 (0.004) | 0.209 (0.003) | 0.293 (0.012) | 0.254 (0.006) | 0.262 (0.000) |
| FSP-RevLM depth: 2 | **0.528 (0.004)** | 0.428 (0.003) | **0.217 (0.006)** | 0.305 (0.013) | 0.243 (0.007) | 0.265 (0.000) |
| FSP-RevLM | 0.524 (0.001) | **0.436 (0.003)** | 0.212 (0.002) | 0.305 (0.006) | **0.255 (0.000)** | **0.277 (0.000)** |

Table 6: **Analysis of future summaries at 3B scale.** We evaluate the effect of different future-summary prediction approaches against vanilla MTP. Results (mean ± s.e. over 3 seeds) report pass@16 for code/math tasks and accuracy for ARC. Handcrafted multi-hot summaries (FSP-BoW) improve over standard MTP, especially on math reasoning (e.g., GSM8K and MATH), while learned summaries (FSP-RevLM) provide the largest gains across math reasoning and ARC tasks.

### D.2.2 EVALUATING PERFORMANCE ON NEXT-TOKEN PREDICTION

Our main comparison evaluates pretrained base models on downstream reasoning benchmarks, since performance on these tasks is a strong indicator for the capabilities we ultimately care about after post-training. For completeness, we also report standard language-modeling metrics, next-token negative log-likelihood (NLL) and next-token accuracy (NTA) on two held-out validation sets, Dolci-Think and Dolmino (Olmo et al., 2025), which are excluded from the pretraining corpus. We compute NLL/NTA over approximately 4M tokens.

The results are shown in Table 7 and Table 8 for the 3B and 8B parameter model respectively. Across methods, we observe that improvements on downstream reasoning benchmarks are substantially larger and more consistent than the differences observed in NLL/NTA, which are often small. Thus auxiliary objectives influence the learned internal representations in a way that improves generalization on downstream tasks, even when the standard next-token prediction metrics remain similar.

| Metric | Dataset | NTP | MTP | DS-MTP | FSP-RevLM |
|---|---|---|---|---|---|
| NLL (lower is better) | Dolci-Think | 3.1813 (0.0000) | 2.9746 (0.0000) | 3.2272 (0.0000) | 2.8012 (0.0000) |
| | Dolmino | 1.0266 (0.0000) | 1.0401 (0.0000) | 1.0239 (0.0000) | 1.0264 (0.0000) |
| NTA | Dolci-Think | 0.4057 (0.0000) | 0.4078 (0.0000) | 0.4046 (0.0000) | 0.4214 (0.0000) |
| | Dolmino | 0.7450 (0.0000) | 0.7457 (0.0000) | 0.7491 (0.0000) | 0.7513 (0.0000) |

Table 7: **Performance on next-token prediction at 3B scale.** We report negative log-likelihood (NLL) and next-token accuracy (NTA) on Dolci-Think and Dolmino. Results are averaged over 3 random seeds (mean ± SE). Lower NLL corresponds to better predictive calibration.

| Metric | Dataset | NTP | MTP | DS-MTP | FSP-RevLM |
|---|---|---|---|---|---|
| NLL (lower is better) | Dolci-Think | 1.3535 (0.0000) | 1.3604 (0.0000) | 1.3571 (0.0000) | 1.3634 (0.0000) |
| | Dolmino | 0.9330 (0.0000) | 0.9386 (0.0000) | 0.9217 (0.0000) | 0.9373 (0.0000) |
| NTA | Dolci-Think | 0.6118 (0.0000) | 0.6123 (0.0000) | 0.6128 (0.0000) | 0.6124 (0.0000) |
| | Dolmino | 0.7652 (0.0000) | 0.7679 (0.0000) | 0.7707 (0.0000) | 0.7696 (0.0000) |

Table 8: **Performance on next-token prediction at 8B scale.** We report negative log-likelihood (NLL) and next-token accuracy (NTA) on Dolci-Think and Dolmino. Results are averaged over 3 random seeds (mean ± SE). Lower NLL corresponds to better predictive calibration.

