# OpenReview forum: "Beyond Multi-Token Prediction: Pretraining LLMs with Future Summaries"
_ICLR.cc/2026/Conference — ICLR 2026 Poster_

### Official Review · Reviewer_WozN · 2025-10-22

**Soundness:** 3
**Presentation:** 3
**Contribution:** 3
**Rating:** 6
**Confidence:** 3

**Summary:**

The work proposes an improved multi-token prediction objective for LLMs. Instead of just predicting the next token, or next few tokens, the method proposes to predict some summary of the future tokens. The summary can either be a simple bag of words set, or a learned embedding from another transformer that encodes the future. They show at the 3B and 8B scale, that future summary prediction outperforms conventional multi-token approaches in a variety of reasoning tasks (ARC, GSM8K, etc.) as well as on some pedagogical tasks.

**Strengths:**

- Interesting idea of predicting a learned summary of the future, instead of raw tokens, to make representations capture more of the future.
- Good evaluations of their multi-token prediction objective against other multi-token prediction baselines, at somewhat large scales (3B-8B) and over serious reasoning benchmarks.
- Paper is generally well written.

**Weaknesses:**

**This method has increased param / training budget, while baselines do not**

This method requires the training of an additional encoder, RevLM, and the experiments do not mention the additional size of this encoder or the computational cost incurred from using it.

 There should be some baseline that is fair in the number of training flops / parameters, since your method gets access to an additional encoder. For example, you should train the MTP baseline with the same number of parameters / flops as your method.


**Learned Future Summary is still susceptible to failure modes**

It's also not clear to me how the learned future summary is "adaptive", in the sense that the future summary embedding is being trained on a "fixed" objective of previous token prediction, given the future. I suspect it's possible to come up with some adversarial sequence for the RevLM to destroy its representation.

Here's one speculative idea. Imagine we also have a stargraph where the paths lead to other stargraphs.  The forward LM starts in the origin of the 1st stargraph, and the RevLM starts at the origin of another stargraph at the end of the path of the first stargraph. Then the RevLM will learn a shortcut solution for predicting the previous token, and making the learned future summary useless.

### Minor
Sibling discovery task: the evaluation metric is vague, where the method's convergence speed is shown to be faster than NTP. But what is the definition of a converged method, is there a raw metric like accuracy score? The relative convergence speed to NTP convergence is not useful if NTP is bad, for example.

**Questions:**

Aside from my concerns above,

Is there a connection between this objective and JEPA / SSL approaches, where we are trying to enforce similarity between a view of the context, and a view of the future?

---

> ### Author Response · Authors · 2025-11-23
>
> Thank you for the thoughtful review and for highlighting the strengths of our work — in particular, the usefulness of predicting a learned future summary, the clarity of the writing, and the strong empirical comparisons at the 3B–8B scale. We appreciate your constructive feedback and address all concerns below.
>
> ---
>
> ## Summary of Concerns and Responses
>
> | Reviewer Concern | Clarification / New Evidence | Action |
> |------------------|-----------------------------|--------|
> | **C1. Additional params / compute vs. baselines** | Identical model size for reverse encoder; training FLOPs increased but justified as the reverse teacher cost is amortized. | A1 Add this discussion in Section 4. |
> | **C2. Learned summary may fail adversarially** | Agree; reverseLM is not perfect but far more principled and robust than token-level heuristics. Triggering such adversarial cases is harder.| A2.2 Add short discussion on remaining open challenges. |
> | **C3. Convergence metric unclear** | All methods reach perfect final score; convergence speed is the meaningful differentiator. | A3.1 Clarify metric definition in Sec. 3.2. |
>
>
>
> ---
>
> ## 1. Parameter / Training compute concerns
>
> > This method requires the training of an additional encoder, RevLM, and the experiments do not mention the additional size of this encoder or the computational cost incurred from using it.
>
>
> Thanks for pointing this out! In our experiments, the reverse model is the same size as the forward model (and other baselines), and it is trained for the same number of steps. As a result, FSP-RevLM roughly doubles the total compute cost compared to standard NTP training. We will update the draft with these details.
>
> > There should be some baseline that is fair in the number of training flops / parameters, since your method gets access to an additional encoder.
>
> Following standard practice in distillation, we do not perform iso-compute comparisons that include the teacher model’s (ReverseLM) cost in the reported compute budget. In practical scenarios, once the ReverseLM is trained it can be used to train student models of different sizes, either smaller or even larger than the teacher. This is consistent with how distillation is typically applied. As a result, the computational cost of training the teacher model is generally amortized and can be treated as a one-time overhead that is excluded from comparisons of student models [1,2]
>
> **Note.** On a separate note, we wish to highlight that FSP-BCE, another contribution of our paper, enables matched comparisons with baseline methods and outperforms both NTP and MTP on several tasks at the 8B scale (Table 3). For the convenience of the reader, we also reproduce the corresponding results from the main paper here.
>
> | Method                     | MBPP                     | GSM8K                    | MATH                     | HumanEval+               | ARC-Challenge             | ARC-Easy                 |
> |----------------------------|---------------------------|---------------------------|---------------------------|---------------------------|----------------------------|---------------------------|
> | NTP                        | 0.657 (0.004)             | **0.716 (0.003)**         | **0.342 (0.008)**         | 0.478 (0.019)             | 0.531 (0.000)              | 0.718 (0.000)             |
> | MTP                        | **0.672 (0.008)**         | 0.678 (0.007)             | 0.309 (0.006)             | **0.541 (0.011)**         | 0.552 (0.000)              | 0.736 (0.000)             |
> | FSP-BCE τ=12               | 0.669 (0.005)             | 0.699 (0.006)             | 0.331 (0.016)             | 0.508 (0.005)             | **0.562 (0.000)**          | **0.737 (0.000)**         |
> | FSP-BCE τ=100              | **0.671 (0.002)**             | **0.714 (0.009)**             | 0.331 (0.007)             | 0.500 (0.019)             | 0.459 (0.000)              | 0.662 (0.000)             |
>
>
> **BCE vs. NTP.**
>
>
> BCE delivers clear gains over NTP on the ARC suite, with ARC-Challenge jumping from 53.1\% → 56.2\% and ARC-Easy from 71.8\% → 73.7\%, and also improves on HumanEval+. NTP only holds its lead on GSM8K and MATH.
>
> **BCE vs. MTP.**
>
> BCE outperforms MTP on several benchmarks, with improvements on ARC Challenge (55.2\% → 56.2\%) and strong gains on Math (30.9\% → 33.1\%) and GSM8k (67.8\% → 71.4\%).

---

> ### Author Response · Authors · 2025-11-23
>
> ## 2. Potential failure modes of future summary prediction
>
>
> > It's also not clear to me how the learned future summary is "adaptive",
>
> We clarify that by “adaptive,” we mean that the model can learn which aspects of the future are relevant for a given prefix, rather than relying on a fixed hand-designed rule. In MTP, the lookahead is predetermined (for example, always the next $k$ tokens), regardless of the input context. Similarly, the bag-of-words summary applies a uniform BCE re-weighting scheme that does not vary with the prefix. In contrast, the reverse LM produces a representation that depends on the input prefix, allowing the learned summary to emphasize different future information depending on the context. This is the sense in which the summary is adaptive.
>
> > I suspect it's possible to come up with some adversarial sequence for the RevLM to destroy its representation. Here's one speculative idea...
>
> We agree that, in adversarially constructed cases, a reverse LM could also learn shortcuts or produce a noisy summary. We do not claim that RevLM is the optimal robust future-summary encoder. Our main argument is that heuristics used in MTP (next-k, random-k [3,4], jump-k [5]) break fundamentally under long horizons or noisy futures. Hence, a learned summary provides a more principled alternative. FSP-RevLM is simply a first step in that direction, and it already delivers strong gains on real-world reasoning benchmarks, which are the settings we ultimately care about. Designing even better learned summaries remains an open and important direction. While failure modes exist in principle, they are significantly harder to trigger than with the proposed bag-of-words or random-lookahead MTP methods [3, 4].
>
> ---
>
> ## 3. Convergence speed metric in sibling discovery
>
> The reviewer raises a fair question about the metric. To clarify, as discussed in Sec. 3.2 (line 295), NTP can eventually learn the task and generate coherent sequences. However, it is highly sample-inefficient because it factorizes the underlying joint distribution in a suboptimal way. All methods eventually reach perfect consistency at convergence (line 310), so comparing time to convergence is a reliable metric.  We will make this explicit in the revised manuscript.
>
> ---
>
> ## 4. Connection to JEPA / SSL
>
> Thanks for this interesting question! There is a loose conceptual analogy in that both JEPA/SSL methods and our approach predict a representation of content that is withheld from the input. Beyond that, however, the objectives diverge substantially. JEPA-style models match continuous representations of alternate views of the same input and are explicitly non-autoregressive: they avoid token-level prediction and do not perform generative modeling. In contrast, FSP-RevLM is designed for autoregressive language modeling, where the goal is to improve next-token prediction by providing the model with a compact summary of the future token sequence. The predicted summary is used only as training-time auxiliary supervision, and generation at inference remains purely next-token autoregressive. Thus, while the high-level idea of predicting a latent representation is shared, the learning objective and generative implications of JEPA and FSP-RevLM are fundamentally different.
>
>
> ---
>
> ## Final Remarks
>
> We again thank the reviewer for their time and raising important questions. We hope that the above clarifications are helpful and would be happy to elaborate further on any of the point. Should these satisfactorily address the concerns, we would be grateful if the reviewer could reassess the work more positively.
>
>
> *References*
>
> [1] "Gemma 2: Improving Open Language Models at a Practical Size" https://arxiv.org/abs/2408.00118
>
> [2] "Gemma 3 Technical Report" https://arxiv.org/abs/2503.19786
>
> [3] "Looking beyond the next token." https://arxiv.org/html/2504.11336v2
>
> [4] "Multi-Token Prediction Needs Registers." https://arxiv.org/abs/2505.10518
>
> [5] "L-MTP: Leap Multi-Token Prediction Beyond Adjacent Context for Large Language Models" https://arxiv.org/abs/2505.17505

---

### Official Review · Reviewer_YWeo · 2025-10-27

**Soundness:** 3
**Presentation:** 3
**Contribution:** 3
**Rating:** 6
**Confidence:** 3

**Summary:**

This paper proposes Future Summary Prediction (FSP) to address the limitations of next-token prediction (NTP) and multi-token prediction (MTP). The authors introduce two variants of FSP and conduct experiments on 3B and 8B models, demonstrating consistent improvements over NTP and MTP across mathematical reasoning, general reasoning, and coding benchmarks.

Key Reasons:
1. The paper presents a well-motivated and principled pretraining objective that addresses genuine limitations of existing approaches.
2. The interpretability of the learned representations and the computational cost of the approach are insufficiently analyzed.

Supporting Arguments

FSP-RevLM achieves consistent gains across multiple benchmarks, validating the empirical effectiveness of the method. However, the underlying mechanism—why future summary prediction improves model performance—remains underexplored.

**Strengths:**

The proposed FSP shifts the modeling focus from predicting individual tokens to capturing the global structure of future content—a fresh and principled direction for language modeling.
Experiments at the 3B and 8B scales are comprehensive, with detailed comparisons against NTP, MTP, and DeepSeek-MTP across multiple benchmarks.

**Weaknesses:**

Although FSP-RevLM achieves strong empirical results, the learned future representations lack interpretability, making it unclear what specific future information is being encoded.
The computational cost of training the reverse language model is not discussed in detail. The paper does not include comparisons of training time or resource consumption.

**Questions:**

1. What is the computational overhead of training the reverse language model? Can it be jointly optimized with the main model?
2. Have you considered applying FSP to other popular open-source LLM architectures?
3. Why does FSP-RevLM slightly underperform NTP on GSM8K? Have you analyzed task-specific failure cases?
4. Do you plan to evaluate FSP on larger models (e.g., 70B) or alternative architectures (e.g., MoE, RWKV)?
5. Is there a way to interpret or visualize the learned future summaries from RevLM to better understand the captured information?

---

> ### Author Response · Authors · 2025-11-23
>
> Thank you for the thoughtful and constructive review. We are glad that you found FSP to be a well-motivated and principled approach that *addresses genuine limitations of existing methods*, and that you appreciated the comprehensive 3B–8B evaluations across reasoning, math, and coding benchmarks. We address all concerns below.
>
> ---
>
>
> ## Summary of Concerns and Responses
>
> | Reviewer Concern | Clarification / New Evidence | Action |
> |------------------|-----------------------------|--------|
> | **C1. Compute and parameters not discussed** | Inference params identical; training FLOPs increased but justified in the current scaling paradigm  | A1.1 Add compute discussion in Sec. 4. |
> | **C2. Lack of interpretability of learned summaries** | Agree; visualization is important, and discuss attention matrix visualization for sibling discovery task. | A2.1 Add note in Sec. 3.2 |
> | **C3. Scaling to 70B / MoE** | Deepseek-v3 scales MTP to 670B and MoE architecture |  |
> | **C4. GSM8K slight drop** | Likely due to noisy summary signal; still strong overall gains. |  |
>
> ---
>
> ## 1. Computational Overhead of Reverse LLM
>
> > What is the computational overhead of training the reverse language model? Can it be jointly optimized with the main model?
>
> Thanks for raising this important question! In our implementation, the reverse language model is trained for the same number of steps as the forward model (and all other baselines). Consequently, FSP-RevLM approximately doubles the total training compute relative to standard NTP. We will clarify this in the updated draft.
>
> Regarding the reviewer’s suggestion about joint optimization: we agree this is a promising direction. In principle, training the forward and reverse models jointly should reduce the computational overhead, and exploring such coupled training schemes is indeed compelling future work.
>
> Finally, while FSP-RevLM increases training FLOPs, we believe this tradeoff is reasonable in today’s compute-rich, data-limited scaling regime. The field has effectively hit the data wall, whereas available compute continues to grow. Progress increasingly depends on using this growing compute-per-token budget to extract more value from fixed datasets. In this context, methods that deliver measurable gains without requiring additional data, even at higher compute cost, are valuable.
>
> ---
>
> ## 2. Interpretability of Learned Future Summaries
>
> > Is there a way to interpret or visualize the learned future summaries from RevLM to better understand the captured information?
>
> Thank you for raising this important point. We agree that interpreting the learned future summaries is crucial for understanding how RevLM captures information from future context.
>
> For the sibling-discovery experiment (Section 3.2), we conducted a preliminary analysis using the attention weights from the final layer, which directly contribute to the future-summary representation produced by RevLM. We observe that queries associated with a particular component in the sequence attend predominantly to keys from the same component. In other words, cross-component attention is limited. This suggests that the learned future summaries preserve the informative structure of the future input by focusing on intra-component tokens. We will include visualizations of these attention patterns in the revised draft.
>
> More broadly, prior work shows that hidden states of LLMs naturally encode detailed information about surrounding tokens (see Section 5 of Physics of LLMs, arXiv:2309.14316), which supports our intuition that reverse-direction representations can capture meaningful long-range patterns. Extending this analysis with additional interpretability tools (linear probing, activation patching, etc.) is a promising direction for future work.

---

> > ### Author Response · Authors · 2025-11-23
> >
> > ## 3. Applicability to Larger (70B) and Other Architectures
> >
> > > Do you plan to evaluate FSP on larger models (e.g., 70B)
> >
> > We currently evaluate FSP at the 3B and 8B scales, where the trends are consistent and clearly positive. While we do not yet have the infrastructure to train a 70B model, there is strong reason to expect favorable scaling. Prior work (Gloeckle et al. [1]) shows that MTP continues to improve at larger model sizes, and MTP-style objectives have been successfully adopted in frontier systems such as DeepSeek-V3 (including MoE variants, scaled upto 670B!). Since FSP-BCE already outperforms DeepSeek-MTP as we scale from 3B → 8B, it is reasonable to anticipate that these gains would amplify at larger scales.
> >
> > A natural next step is to conduct a scaling-law analysis in the mid-range regime to quantitatively extrapolate FSP performance toward sizes like 70B. We view this as an exciting and important direction for future work.
> >
> > > Have you considered applying FSP to other popular open-source LLM architectures? or alternative architectures (e.g., MoE, RWKV)?
> >
> > Our current experiments are conducted using the Llama architecture [2], specifically its dense variant without Mixture-of-Experts (MoE) components. Exploring FSP on alternative architectures is a natural direction for future work. In particular, applying FSP to MoE models or to non-Transformer architectures such as RWKV would provide valuable insight into how the method behaves under different computational and inductive-bias regimes.
> >
> > ---
> >
> > ## 4. GSM8K Underperformance: Task-Specific Analysis
> > > Why does FSP-RevLM slightly underperform NTP on GSM8K? Have you analyzed task-specific failure cases?
> >
> > This is an excellent question. To be fully transparent, we do not yet have a definitive explanation for why FSP-RevLM slightly underperforms NTP on GSM8K at the 8B scale. Pinpointing task-specific failure modes is inherently challenging, especially because GSM8K may be particularly sensitive to the quality of the reverse-LM summary. One plausible hypothesis is that, for some problems, the reverse model may generate less informative summaries, effectively injecting noise into training.
> >
> > That said, it is important to emphasize that across the full evaluation suite, FSP-RevLM delivers consistent and robust improvements, and the GSM8K dip appears to be an isolated case rather than a systemic issue. We also note that at the 3B scale, FSP-RevLM outperforms NTP on GSM8K (Table 2), which suggests that the discrepancy may be related to scale or training dynamics rather than a fundamental limitation of the method.
> >
> > Improving the design and training of future summary encoders, including more reliable reverse-LM objectives, is an important direction for immediate follow-up work.
> >
> > ---
> >
> > ## Final Remarks
> >
> > Thank you again for the careful review and for recognizing the strengths of our approach. We hope that the above clarifications are helpful and would be happy to elaborate further on any of the point. Should these satisfactorily address the concerns, we would be grateful if the reviewer could reassess the work more positively.
> >
> >
> >
> > *References*
> >
> > [1] "Better & Faster Large Language Models via Multi-token Prediction" https://arxiv.org/abs/2404.19737
> >
> > [2] "LLaMA: Open and Efficient Foundation Language Models" https://arxiv.org/abs/2302.13971

---

### Official Review · Reviewer_QvQ8 · 2025-10-31

**Soundness:** 3
**Presentation:** 3
**Contribution:** 2
**Rating:** 2
**Confidence:** 4

**Summary:**

This paper presents two methods for improving transformer training.  One predicts the future bag-of-words with an additional head (FSP-BCE) and another predicts the hidden representation of a backwards running transformer (FSP-RevLM).  There are some results on path-star, sibling discovery, and 8b scale pre-training.  I think this is a nice topic, but I see a few issues with the paper.  First, the presentation of two very different methods gives the papers a somewhat disjointed feel.  The advantages and disadvantages of each method feel like they are not discussed enough.  For example, on some datasets, FSP-BCE will converge to predicting a marginal distribution over all the tokens in the dataset, if the sequences are sufficiently long and diverse.  In this case, it seems like it would not be very useful.  It also seems like this the effectiveness of this method will be sensitive to how tokenization is done.  For example, if we used a character-level model, it seems like it would be almost completely useless.  I don't think that's a decisive disadvantage by itself, but a paper focusing on this method exclusively could explore these tradeoffs.

Another serious issue is that I believe that the RevLM method has been proposed before in the literature, (Serdyuk et. al 2017), and while the method here has some differences, it seems like the paper should be written so that the novelties in the proposed method can be properly highlighted.

**Strengths:**

The paper deals with an important topic, which is how to move beyond the next-token prediction objective in a practical way.  I believe that the experiments are also correctly implemented and described in a clear way.  Some of the empirical improvements, especially with FSP-RevLM are also impressive.

**Weaknesses:**

In my opinion, the biggest issue with this paper is that I don't believe that the FSP-RevLM method's novelty is correctly explained with respect to the existing literature.  There is a paper called Twin Networks, published in ICML 2017 (Serdyuk et. al), which I believe is essentially the same approach, of predicting the hidden state of a backward-trained RNN.  Now, there may be a difference here, in that this paper uses transformers instead of RNNs.  However, those differences should be discussed and this paper should be cited.  I found another paper, not cited in this work, called "ProphetNet" from (Qi et. al) in 2020 which predicts future n-grams, which also seems quite closely related to the bag-of-words prediction in this paper.    The z-forcing paper (Goyal et. al 2017) also predicts the hidden states of a backwards running RNN.  See section 3.3 in the arxiv paper.  The fact that this very relevant and fairly old work is not discussed, gives me concerns.

It's less important, but there is a recent paper "Joint Token Prediction" (Ahn et. al 2025) which deals with learning belief states using a multi-token prediction method.  However, I am willing to accept this not being cited due to the work being quite recent.

Another issue I have is that FSP-BCE seems like it could be sensitive to the horizon that you use, depending on how the dataset is constructed.  Perhaps empirically it's not an issue, but it seems like you can run into trouble with method if the sequences become long enough, and eventually the distribution over future tokens become similar (for example if our dataset is drawn from one very long sequence, then this would be an issue).  However, if the dataset splits all the sequences into different documents, perhaps this won't be an issue.

**Questions:**

The paper has experiments on the "path star" task, but why didn't you also run on the more general (and much harder) star graph task?

What's the added computational cost of training the RevLM model on the right-to-left sequences?  Presumably it's twice the number of total parameters used during training and a little over twice the cost to train?

---

> ### Author Response · Authors · 2025-11-23
>
> Thank you for the thoughtful and constructive review. We are glad that you found the improvements with FSP-RevLM to be impressive. We address all of your concerns below.
>
> ## Summary of Concerns and Responses
>
> | Reviewer Concern | Clarification / New Evidence | Action |
> |------------------|-----------------------------|--------|
> | **C1. Similarity to prior works: TwinNet / z-Forcing / ProphetNet** | Our key contributions are beyond just the method FSP-RevLM. We propose a unified framework that spans a variety of techniques around MTP, highlighting key limitations and the  need for principled future summarization approaches.  | A1.1 Add relevant discussion in the related works |
> | **C2/C3. Advantages of FSP-BCE and FSP-RevLM and disjoint writuep** | We will strengthen the narrative: FSP-BCE shows inherent limits especially under long future → motivates adaptive learned summaries. | A2.1 Add connecting text in Introduction and Sec. 3.2. |
> | **C4. Compute cost of training RevLM** | ~2× NTP cost; acceptable given data wall + rising compute budgets. | A5.1 Add compute note in Sec. 4. |
>
> ## 1. Comparison with prior works in the RNN literature
>
> We thank the reviewer for raising this important point, and we agree we should have cited these highly relevant works from the RNN literature. We will update the discussion on related works with emphasis on twin networks and the other relevant works mentioned.
>
> > Comparison with Twin Networks (Serdyuk et. al)
>
> We appreciate the reviewer pointing that the high level idea behind our reverse LM objectives is not entirely new, and was explored with RNNs in Twin Networks. Below, we clarify the key technical differences and our contributions beyond that work.
>
> **a) Our contributions go beyond just the FSP-RevLM algorithm**
>
> Future summary prediction provides a unified view of a broad family of pretraining objectives, covering prior multi-token prediction (MTP), variants that sample random future tokens [1, 2], and our proposed bag-of-words (FSP-BCE) and reverseLM (FSP-RevLM) summary prediction objectives.
>
> This framework clarifies the tradeoffs of each objective. For example, FSP-BCE directly addresses the path–star graph problem, where NTP and MTP struggle to aggregate long-range, unordered future information. But FSP-BCE is still suboptimal when only part of the future is relevant, as shown in the sibling-discovery task. These insights motivate using a reverse LM to compute a learned, task-adaptive future summary.
>
> Overall, our contribution is not limited to a specific auxiliary loss, but the broader perspective that future summary prediction provides for understanding and designing pretraining objectives, and *the reasoning and evidence that show when simpler approaches fail and why a learned summary with reverseLM becomes necessary.*
>
> **b) Scaling to transformers is non trivial and common in the community**
>
> Just as TwinNet anticipated aspects of our reverse LM component, earlier work also explored related forms of multi-token prediction. This includes approaches that train models to predict multiple future tokens or parallel token blocks from a given prefix [4, 5], well before Idrissi et al. [3]. These efforts did not establish MTP as a broadly effective pretraining objective for large-scale Transformers, but they do show that the underlying intuition has been around for some time. Idrissi et al. [3] deserves credit for identifying a formulation of MTP that works in modern LLM training, and our work should be understood in the same spirit. We extend this line of evidence by showing that both MTP and future summary prediction can be integrated cleanly into Transformer pretraining, scaled up, and shown to deliver robust gains.
>
> Also, it is worth noting that there are several well accepted works that have brought RNN-style inductive biases into transformers. Looped Transformers [6] add iterative refinement over hidden states, effectively introducing a recurrence-like update reminiscent of RNN state transitions. The Pause Token mechanism [7] allows the model to update its internal state over multiple steps without emitting a token, again echoing recurrent processing. These works illustrate that ideas from earlier sequence models continue to be revisited and reinterpreted within the Transformer framework and modern LLM settings, and our reverse LM summary objective fits naturally within this trajectory.

---

> ### Author Response · Authors · 2025-11-23
>
> (continued from the previous comment)
>
> > Comparison with z-forcing (Goyal et al.)
>
> Thank you for highlighting this work. While most of the arguments from the above response follow, the main technical distinction relative to our setting is that they use a reverse RNN to infer latent variables, which are then fed into the forward RNN to compute its hidden states. In our approach, the reverse LM is not performing latent-variable inference. It is used only to generate targets for an auxiliary prediction head. Moreover, unlike z-forcing, our method does not require the reverse LM at inference time. The reverseLM is purely a training-time component, whereas z-forcing relies on the reverse RNN during inference as well.
>
>
> > Comparison with ProphetNet (Qi et al.)
>
> Thanks for raising this point. The proposed FSP-BCE differs in several important ways from the future n-gram prediction approach used in ProphetNet. ProphetNet applies a per–future-token cross-entropy loss and relies on relative positional encodings within its shared N-stream self-attention layer. As a result, it still computes and supervises an explicit loss for each future token.
>
> In contrast, FSP-BCE operates on a summary vector that discards positional information. Instead of supervising each future token individually, it uses a (weighted) binary cross-entropy objective over the unordered set of future tokens. This leads to a structurally different training signal that captures future content without enforcing per-token positional predictions.
>
> > Comparison with Joint Token Prediction (Ahn el al.)
>
> We thank the reviewer for pointing this recent work and we will cite it in the updated draft.
>
> ---
>
>
> ## 2. Disjoint feel in the presentation of two methods
> > First, the presentation of two very different methods gives the papers a somewhat disjointed feel.
>
>
> We agree that the connection may not have been emphasized strongly enough in the introduction, and we will update the manuscript accordingly.
>
> A key contribution of our paper is the introduction of a unified framework that uses a single auxiliary head (as opposed to multiple heads, as in MTP) to capture long-range future dependencies. Within this framework, both the bag-of-words summary (FSP-BCE) and the learned summary produced by the reverse language model (FSP-RevLM) are two  instantiations of the same framework.
>
> Moreover, Section 3.2 explicitly discusses the limitations of the bag-of-words summary, as well as the broader class of hand-crafted heuristics it represents. Our analysis with the sibling discovery task shows that predicting all future tokens as in FSP-BCE, is fundamentally limited in situations where only part of the future is relevant. This analysis directly motivates the need for more principled mechanisms for extracting the salient aspects of the future. FSP-RevLM is precisely a step in that direction, and it further reinforces the connection between the approaches within our proposed framework.
>
> ---
>
> ## 3. Advantages/Disadvantages of each method
> > The advantages and disadvantages of each method feel like they are not discussed enough. ....on some datasets, FSP-BCE will converge to predicting a marginal distribution over all the tokens in the dataset, if the sequences are sufficiently long and diverse. In this case, it seems like it would not be very useful
>
> > Another issue I have is that FSP-BCE seems like it could be sensitive to the horizon that you use,  .....
>
> We agree that, in principle, a bag-of-words summary can lose informativeness when the prediction horizon becomes extremely large, for example when drawing from a single very long continuous sequence where the future-token distribution eventually flattens. In practice, however, this issue is minimal. Most pretraining corpora consist of many short, independent documents, so future-token distributions reset frequently and maintain meaningful variation. Empirically, even relatively long windows (for example, τ = 100) continued to yield strong gains at the 8B scale (Table 3).
>
> We also reiterate that Section 3 provides a detailed analysis of the advantages and disadvantages of FSP-BCE. In Section 3.1, we show that it effectively solves the path–star graph task by efficiently modeling long-range future content. However, when not all future information is relevant, as in the sibling-discovery task, FSP-BCE faces fundamental limitations.
>
> These failure modes were a core motivation for moving beyond hand-crafted summaries. The reverseLM-based summary addresses these issues by adaptively focusing on the important parts of the future context. Rather than treating all future tokens as equally relevant, it produces a learned embedding that is more robust when the future is long, noisy, or contains unpredictable content.

---

> ### Author Response · Authors · 2025-11-23
>
> ## 4. Compute concerns
> > What's the added computational cost of training the RevLM model on the right-to-left sequences? .....
>
> Yes, FSP-RevLM roughly doubles the compute cost relative to standard NTP training. However, we believe this tradeoff is justified in the current scaling regime. The field has effectively reached the data wall, while available compute continues to grow. Future progress will depend on how effectively we can use this increasing compute-per-token to extract more value from the data we already have, even if doing so requires higher training-time compute.
>
> Further, note that all approaches — NTP, MTP, and the proposed FSP-RevLM — use exactly the same number of parameters at inference. The reverse LM encoder is used *only during training*. *Thus, performance gains are not due to increased capacity at inference.*
>
> ---
>
> ## 5. Other concerns
> > ....if we used a character-level model, it seems like it would be almost completely useless
>
> A character-level tokenization would perform worse for standard language-modeling objectives such as NTP itself. We do not see this as a relevant concern, since the community has largely converged on byte-pair encoding (BPE) for tokenization modern LLMs.
>
>
> > The paper has experiments on the "path star" task, but why didn't you also run on the more general (and much harder) star graph task?
>
> Our primary goal with the path star task was to highlight the failure mode of previous multi-token objectives, that scale the number of auxiliary heads to predict more future tokens. Hence, we demostrated it via minimal examples from the path-star graph task.
>
>
> ---
>
> ## Final Remarks
>
> We again thank the reviewer for their time and for the thoughtful and detailed feedback. We hope the clarifications above address their concerns, and we would be glad to elaborate further on any of the points. If these responses satisfactorily resolve the raised issues, we would greatly appreciate a more positive reassessment of the work.
>
> *References*
>
> [1] "Looking beyond the next token." https://arxiv.org/html/2504.11336v2
>
> [2] "Multi-Token Prediction Needs Registers." https://arxiv.org/abs/2505.10518
>
> [3] "Better & faster large language models via multi-token prediction." https://arxiv.org/abs/2404.19737
>
> [4] "Image captioners are scalable vision learners too."
> https://arxiv.org/abs/2306.07915
>
> [5] "Pass: Parallel speculative sampling." https://arxiv.org/abs/2311.13581
>
> [6] "Looped transformers as programmable computers." https://arxiv.org/abs/2301.13196
>
> [7] "Think before you speak: Training language models with pause tokens." https://arxiv.org/abs/2310.02226

---

### Official Review · Reviewer_r6NQ · 2025-11-01

**Soundness:** 4
**Presentation:** 4
**Contribution:** 4
**Rating:** 8
**Confidence:** 5

**Summary:**

The authors propose a novel pretraining objective called Future Summary Prediction (FSP). Instead of predicting specific future tokens, FSP trains an auxiliary head on the LLM to predict a single, compact summary of the long-term future of a sequence. This forces the model to develop a more global understanding and plan ahead, significantly reducing its reliance on myopic, shortcut-based learning.

FSP augments the standard NTP loss with an auxiliary loss that predicts a summary of a long future window. The paper explores two ways to create this future summary:
Hand-crafted Summary (FSP-BCE): This is a simple but effective approach where the summary is a "bag-of-words" (a multi-hot vector) of all unique tokens that will appear in a long future window. The model is trained to predict the presence of these future tokens, without needing to know their exact order or position. This forces the model to look far ahead, making it much harder to rely on simple shortcuts.
Learned Summary (FSP-RevLM): This is the paper's main and most powerful contribution. It addresses a weakness in handcrafted summaries, which can be "noisy" by including irrelevant future information. A separate reverse language model (RevLM) is trained to read sequences from right to left.The hidden state of this RevLM, after it has processed the future part of a sequence, serves as a rich, compact, and adaptive summary of that future. The main (forward) model is then trained to predict this learned summary vector. The RevLM naturally learns to emphasize what is important and predictable about the future, filtering out noise.

**Strengths:**

Novelty: This paper proposes a new dimension towards thinking about multi-token prediction in LLMs
Superior Performance on Benchmarks: At the 8B scale, FSP-RevLM consistently outperforms NTP, MTP, and DeepSeek-MTP across a range of benchmarks. It shows significant improvements on:
Reasoning: Up to a 5% absolute gain on ARC (AI2 Reasoning Challenge).
Math: A 4.2% gain on MATH and 3.5% on GSM8K compared to standard MTP.
Coding: Achieves the highest score on MBPP and is competitive on HumanEval+.

**Weaknesses:**

The speedup due to this change is not mentioned in the main paper.

**Questions:**

1. Does this technique only boost model accuracy or do you see any boost in inference speed per token by generating multiple tokens at a time?

---

> ### Author Response · Authors · 2025-11-23
>
> We thank the reviewer for their positive and insightful feedback! We are glad they found our work novel and that it *proposes a new dimension towards thinking about multi-token prediction in LLM*, along with good performance across benchmarks. We now address the reviewer’s concerns below.
>
> > Does this technique only boost model accuracy or do you see any boost in inference speed per token by generating multiple tokens at a time?
>
> Our work is primarily aimed at improving the quality of learned representations by providing the model with a richer and more future-aware training signal. Because the auxiliary head in FSP predicts summaries of future content rather than future tokens themselves, it cannot be used directly at inference time, and therefore does not provide inference-time speedups in its current form.
>
> In principle, one could train multiple auxiliary heads on top of the learned FSP representation, for example through a brief phase of continued pretraining, and then use these heads to generate several tokens concurrently during inference. Since the FSP representation is more future aware, it may support higher acceptance rates and more stable multi-token generation than existing MTP-style approaches. Exploring such extensions of our framework is promising, but falls outside the scope of the present work.

---

### Author Response · Authors · 2025-11-26
**Summary of Changes to the Draft**

We summarize the new sections added to the paper (highlighted in red color) to address the reviewers comments. We believe we have thoroughly addressed all the concerns raised by the reviewers. We wish to check if our responses have sufficiently addressed their concerns. If so, we would be grateful if they could update your score accordingly.

- **Section 3**: Addressing disjoint feeling of two methods (Reviewer QvQ8)

→ We provide justification how the future summary prediction framework presents a unified view of the various pretraining objectives and how it helps understands their relative advantages/disadvantages

- **Section 3.2**: Justifying the convergence speedup metric (Reviewer WozN)

→ We specify in lines 320-321 that since NTP achieves perfect consistency score at convergence, hence time to convergence is a reliable metric for comparison.

- **Section 4.1**: Missing details regarding implementation & computational cost of FSP-RevLM (Reviewer QvQ8, YWeo, WozN)

→ We have added details regarding our experiment design being iso-data (and not iso-compute) and the details regarding the implementation of FSP-RevLM

- **Section 5**: Missing relevant prior works on RNNs (Reviewer QvQ8)

→ We provide comparison with JTP (Ahn et al., 2025) and ProphetNet (Qi et al., 2020) in the paragraph *Going Beyond Immediate Next Token*, and comparison with Twin Networks (Serdyuk et al., 2017) and z-forcing (Goyal et al., 2017) in the paragraph *Bidrectional models*.

- **Section B.2**: Interpretability of the learned future summaries (Reviewer YWeo)

→ In Figure 6, we present the results of our preliminary analysis using the attention weights from the final layer, for the sibling-discovery task.

---

### Author Response · Authors · 2025-11-26
**General Response**

We thank the reviewers for their thoughtful feedback and constructive questions. We are encouraged by their recognition of the significance, novelty, and technical rigor of our work.


- **Novel and meaningful direction beyond NTP/MTP**


→  “A new dimension towards thinking about multi-token prediction in LLMs.” — Reviewer r6NQ

→ “Shifts the focus to capturing the global structure of future content—a fresh and principled direction.” — Reviewer YWeo

→ “A well-motivated objective that addresses genuine limitations of existing approaches.” — Reviewer YWeo

- **Strong and consistent empirical gains at scale**

→ “FSP-RevLM consistently outperforms NTP, MTP, and DeepSeek-MTP… up to a 5% gain on ARC.” — Reviewer r6NQ

→ “Good evaluations… at 3B–8B over serious reasoning benchmarks.” — Reviewer WozN

→ "Some of the empirical improvements, especially with FSP-RevLM are also impressive." — Reviewer QvQ8

- **Well-executed and clearly presented experiments**

→ “Experiments are correctly implemented and described in a clear way.” — Reviewer QvQ8

→ “Paper is generally well written.” — Reviewer WozN

----

We also address here the shared concern of reviewer QvQ8, YWeo, and WozN on the implementation and computational cost of training the reverse model in FSP-RevLM.

We clarify three important points:

**(i) Inference-time parameters remain identical.**
All approaches — NTP, MTP, and our FSP-RevLM — use exactly the same number of parameters at inference. The reverse LM encoder is used *only during training*. Thus, performance gains are not due to increased capacity at inference.

**(ii) Training compute increases, but this is acceptable and aligned with the current scaling regime.**
Yes, FSP-RevLM introduces additional training FLOPs.  However, we believe this tradeoff is acceptable in the current scaling regime. The field has effectively hit the data wall, while available compute continues to grow. The future progress hinges on how do we use this ever growing compute-per-token to more effectively model the available data, even if it comes at a higher compute cost. Thus, any gains, while using the same data are pretty important even if they come at an additional comptue cost.

**(iii) Regarding compute matched baselines**

All experiments are conducted under iso-data conditions, which means that all methods are trained on identical datasets. For the proposed FSP-RevLM, this implies that both the forward and reverse models are trained on the same data. Following standard practice in distillation, we do not perform iso-compute comparisons that include the teacher model’s (ReverseLM) cost in the reported compute budget. In practical scenarios, once the ReverseLM is trained it can be used to train student models of different sizes, either smaller or even larger than the teacher. This is consistent with how distillation is typically applied. As a result, the computational cost of training the teacher model is generally amortized and can be treated as a one-time overhead that is excluded from comparisons of student models [1,2]


*References*

[1] "Gemma 2: Improving Open Language Models at a Practical Size" https://arxiv.org/abs/2408.00118

[2] "Gemma 3 Technical Report" https://arxiv.org/abs/2503.19786

---

### Meta-Review · Area_Chair_MtkA · 2026-01-05

**Summary:**

The paper introduces an alternative training objective to next-token or multi-token prediction for large language models. The proposed approach consists of predicting a summary of future tokens, which can take different forms. Two instances are explored: (i) a bag-of-words representation of future tokens, and (ii) an embedding produced by a separate model that encodes the future text in a backward manner. Experiments conducted with 3B and 8B models show consistent improvements over multi-token prediction baselines on a range of reasoning tasks.

The reviewers acknowledge the importance of the problem and the empirical gains over next-token baselines. They request several clarifications, including a comparison of the computational cost with respect to baselines and a deeper analysis or interpretability of the observed improvements. One reviewer raises substantial concerns regarding the similarity of the proposed approach to prior work, in particular earlier models—often based on RNNs—that also leverage predictions of future representations. This constitutes the main criticism of the paper.

**Reviewer Concerns:**

In their rebuttal, the authors address the reviewers’ questions and acknowledge the conceptual similarities with prior work, providing a more detailed positioning of their contribution with respect to existing approaches. While the general idea can indeed be traced back to earlier contributions, the specific formulation, implementation, and empirical evaluation in the context of modern transformer-based LLMs differ in meaningful ways. Overall, the contribution is relatively incremental, but the proposed idea is simple, well motivated, and empirically effective, leading to consistent improvements for the small- and medium-scale models evaluated.. In my view, the authors have satisfactorily addressed the main criticisms raised by the reviewers.

Considering that all reviewers except one were in favor of acceptance, I recommend acceptance.

**Reviewer Scores:**

Rr6NQ, rating 8, will probably keep their score

RQvQ8, rating 2, the authors have answered all the concerns/questions, would probably raise their score

RYWeo, rating 6, all the questions adressed in the rebutal, would probably keep their rating

RWozN, rating 6, again, all questions answered, would probably keep their rating

---

### Decision · Program_Chairs · 2026-01-26

Accept (Poster)